_Article_

# Cyclin-dependent kinase inhibitor p18 regulates lineage transitions of excitatory neurons, astrocytes, and interneurons in the mouse cortex

Wonyoung Lee[1], Byunghee Kang [2], Hyo-Min Kim [3], Tsuyoshi Ishida[4], Minkyung Shin[1], Misato Iwashita[1], Masahiro Nitta[5], Aki Shiraishi[6], Hiroshi Kiyonari [6], Koichiro Shimoya[7], Kazuto Masamoto[5], Tae-Young Roh [8✉] & Yoichi Kosodo [1,9✉]

## Abstract

Neural stem cells (NSCs) can give rise to both neurons and glia, but the regulatory mechanisms governing their differentiation transitions remain incompletely understood. Here, we address the role of cyclin-dependent kinase inhibitors (CDKIs) in the later stages of dorsal cortical development. We find that the CDKIs p18 and p27 are upregulated at the onset of astrocyte generation. Acute manipulation of p18 and p27 levels shows that CDKIs modulate lineage switching between upper-layer neurons and astrocytes at the transitional stage. We generate a conditional knock-in mouse model to induce p18 in NSCs. The transcriptomic deconvolution of microdissected tissue reveals that increased levels of p18 promote glial cell development and activate Delta-Notch signaling. Furthermore, we show that p18 upregulates the homeobox transcription factor Dlx2 to subsequently induce the differentiation of olfactory bulb interneurons while reducing the numbers of upper-layer neurons and astrocytes at the perinatal stage. Clonal analysis using transposon-based reporters reveals that the transition from the astrocyte to the interneuron lineage is potentiated by p18 at the single-cell level. In sum, our study reports a function of p18 in determining the developmental boundaries among different cellular lineages arising sequentially from NSCs in the dorsal cortex.

**Keywords** Cyclin-dependent Kinase Inhibitors; Astrocyte Differentiation; Ink4 Family; Neural Stem Cells; Brain Development
**Subject Categories** Development; Neuroscience; Stem Cells & Regenerative Medicine

## Introduction

One of the astonishing abilities of neural stem cells (NSCs) is that they can differentiate into several subtypes of neurons and macroglial cells, namely, astrocytes and oligodendrocytes, in the developing brain. Fate choice between neuron and glial cells is critical to determining brain structure, function, and evolution. In the neocortex, the production of neurons and glial cells sequentially takes place along the developmental timing (Kriegstein and Alvarez-Buylla, 2009; Taverna et al, 2014). Initially, neurons are produced directly from apical progenitors (neuroepithelial and radial glial cells (RGCs)), the subtypes of NSCs located in the ventricular zone (VZ) maintaining the apical membrane and junctional complex, or via basally located intermediate progenitors. Subsequently, apical progenitors start to produce astrocytes and then oligodendrocytes. Therefore, it is assumed that there is a switching mechanism from neuron to glial production in the differentiation program of NSCs. Indeed, previous studies have proposed that several pathways are involved in switching NSC progenies from neural to glial cells. Toward astrocytic differentiation, in particular, the roles of signaling pathways (gp130/JAK/STAT (Nakashima et al, 1999), MEK/MAPK (Li et al, 2012), and FGF (Dinh Duong et al, 2019)), transcription factors (Sox9 (Kang et al, 2012) and Zbtb20 (Nagao et al, 2016)), and epigenetic regulation (Nakagawa et al, 2020; Tiwari et al, 2018) are considered to have significant contributions.

Another possible prominent factor in deciding the cell fate of NSCs is the length of the cell cycle (Dalton, 2015; Dehay and Kennedy, 2007; Salomoni and Calegari, 2010). It has been well illustrated that the cell cycle length of the apical progenitor, particularly the G1 phase, is prolonged along stages of neural differentiation during brain development (Caviness et al, 1999; Hindley and Philpott, 2012). The length of each cell cycle phase is coordinated by cyclins, cyclin-dependent kinases (CDKs), and their inhibitors (CDKIs). In total, 8 CDKIs exist in the mouse and

[1]Neural Regeneration Lab, Neural Circuit Research Group, Korea Brain Research Institute, Daegu, Republic of Korea. [2]Department of Life Sciences, Pohang University of Science and Technology, Pohang, Republic of Korea. [3]College of Pharmacy, Ewha Womans University, Seoul, Republic of Korea. [4]Department of Obstetrics and Gynecology, Kobe Tokushukai Hospital, Kobe, Japan. [5]Center for Neuroscience and Biomedical Engineering, The University of Electro-Communications, Tokyo, Japan. [6]Laboratory for Animal Resources and Genetic Engineering, RIKEN Center for Biosystems Dynamics Research, Kobe, Japan. [7]Department of Obstetrics and Gynecology, Kawasaki Medical School, Kurashiki, Japan. [8]Department of Life Sciences, Ewha Womans University, Seoul, Republic of Korea. [9]Department of Brain Sciences, Daegu Gyeongbuk Institute of Science and Technology, Daegu, Republic of Korea. ✉E-mail: tyroh@ewha.ac.kr; kosodo@kbri.re.kr

human genomes. While Cdkn3 is expressed mostly in cancer (Nalepa et al, 2013), other CDKIs are categorized into the Cip/Kip family (p21 encoded by *cdkn1a*, p27/*cdkn1b*, and p57/*cdkn1c*) and INK4 family (p16/*cdkn2a*, p15/*cdkn2b*, p18/*cdkn2c*, and p19/*cdkn2d*) (Besson et al, 2008). Among Cip/Kip family members, p27 and p57 are known to regulate several events in neural differentiation, for instance, enhancing neurogenesis and neural migration (Itoh et al, 2007; Mairet-Coello et al, 2012; Nguyen et al, 2006) as well as NSC quiescence in the adult brain (Andreu et al, 2015; Caron et al, 2018; Furutachi et al, 2013). It has also been reported that p57 determines the origin of adult NSCs at the lateral ganglionic eminence (GE) (Furutachi et al, 2015; Harada et al, 2021). In contrast to the intensive analyses of p27 and p57 in the neurogenic phase, their roles in gliogenesis remain elusive. An earlier work by Tury et al described that the overexpression (OE) of p27 and p57 in the embryonic rat brain resulted in ectopic GFAP-positive cells in the postnatal day (P) 10 brain (Tury et al, 2011). However, to uncover the watershed between neural and glial lineages, it is essential to address whether CDKI-induced glial features appear in an immediate manner or later as an indirect consequence. Hence, multi-aspect investigations of neural and glial signatures combined with the acute manipulation of CDKIs in NSCs need to be conducted.

It should also be mentioned that, compared to that of the Cip/Kip family, the role of the INK4 family in brain development has yet to be explored (Ding et al, 2020; Grison and Atanasoski, 2020). Curiously, previous studies using knockout mice have shown that individual p27 (Fero et al, 1996; Kiyokawa et al, 1996; Nakayama et al, 1996) and p18 (Franklin et al, 1998) null mice share phenotypic similarities in the aspects of increased body size with multiple organ hyperplasia even though they belong to different subfamilies. Such a phenotype is exaggerated in p18/p27 double null mice, suggesting that p18 and p27 act in separate pathways to regulate tissue growth collaboratively (Franklin et al, 1998). On the other hand, INK4 family gene double null mice, such as p15/p18 (Latres et al, 2000) and p18/p19 (Zindy et al, 2001), exhibit only minor phenotypes at birth, indicating that they function in different cellular lineages and pathways.

Here, we investigated the role of CDKIs in the later stage of cortical development. Among CDKIs, we observed that p18 and p27 increased in apical progenitors in the VZ at the onset of gliogenesis. Manipulation of CDKI expression revealed their role in lineage switching between neurons and astrocytes. Remarkably, transcriptomic deconvolution using a conditional p18 knock-in mouse and functional molecular analysis elucidated not only the transition from neurons to astrocytes but also further lineage switching toward olfactory bulb (OB) interneurons. These results indicate that p18 integratedly determines the lineage boundaries of progeny sequentially arising from NSCs of dorsal cortical origin.

# Results

## Expression profiling of CDKIs in the neural and glial generation stage

To address the role of CDKIs in apical progenitors in the developing brain, we first investigated the expression levels of all CDKIs by re-examining published single-cell transcriptome data

(La Manno et al, 2021). Among eight CDKIs in the mouse genome classified as *Cdkn1*, *Cdkn2*, and *Cdkn3*, *p18* and *p27* showed a peak at the onset of the gliogenesis stage (E15-16, Fig. EV1A). We then attempted immunostaining using specific antibodies against p18 and p27 (Fig. EV1B,C) to profile the spatiotemporal localization of each protein in the *AldH1l1*-EGFP mouse line, which expresses EGFP as an astrocyte-specific reporter (Fig. EV1D) (Nagao et al, 2016; Yang et al, 2011). In the E15.5 dorsal cortex, the p18 signal was recognized mainly in the VZ, where apical progenitors localize (Fig. 1A). This pattern fits well with p18 mRNA expression (Zindy et al, 1997). On the other hand, p27 was found not only in the VZ but also in the intermediate zone (IZ) and cortical plate (CP) (Fig. 1A), as previously reported (Nguyen et al, 2006). We next addressed the stage-dependent transition of p18, p27, and *AldH1l1*-EGFP signals in the VZ from E13.5 to E17.5 (Fig. 1B,C). Notably, p18 showed the highest immunofluorescent signal at E15.5, while p27 showed an increase along the stages. The *AldH1l1*-EGFP signal appeared in the VZ at E15.5, and then the expression level was sustained. To validate the result, we examined the mRNA expression levels of *p18*, *p27*, and *EGFP* in the VZ by capturing the tissue fraction using laser microdissection (LMD), followed by quantitative PCR (qPCR) of the extracted lysate (Figs. 1D and EV1E). Compared to the immunostaining results, a similar tendency was observed (Fig. 1E), verifying that specific expression patterns of p18 and p27 correlated with the initiation of astrocyte generation.

## Role of CDKIs in lineage switching between neurons and astrocytes

To explore whether there is a functional link between the presence of CDKIs and neural to astrocytic differentiation switching, gene manipulation was attempted. We suppressed p18 and p27 functions using a miRNA-based knockdown (KD) system (Appendix Fig. S1A,B) combined with in utero electroporation (IUE). We found increased incorporation of EdU by p18 and p27 KD, indicating that more cells entered the S-phase (Fig. 2A,C). Notably, decreased expression of *AldH1l1*-EGFP was observed by p18 and p27 KD, and the effect was enhanced by double KD (Fig. 2B,C). Previous studies reported that p27 KD resulted in defects in the radial migration of postmitotic neurons (Itoh et al, 2007; Nguyen et al, 2006). We confirmed the same effect on neural migration by p27 KD, while no significant change was recognized by p18 KD (Fig. 2D,E). To address the role of CDKIs, we chose p18 as a principal target to investigate because of its specific function in apical progenitors but not neurons. We verified that the reduction in astrocytic differentiation was due to p18 KD because the *AldH1l1*-EGFP signal was rescued by overexpression (OE) of human p18 (Appendix Fig. S1C).

Next, we performed gain-of-function analysis by delivering the p18 gene to apical progenitors. Cell cycle inhibition by p18 OE in target cells was confirmed by reduced incorporation of EdU (Fig. EV2A). OE of p18 in the *AldH1l1*-EGFP mouse brain at E15.5 enhanced the expression of *AldH1l1*-EGFP in subsequent stages toward P0 compared to the control (Fig. 2F,G). Additionally, p18 OE cells localized close to the ventricular surface, while control cells migrated out at P0 (Fig. EV2B). The p18 OE cells near the ventricular surface at P0 showed increased expression of astrocyte markers (Sox9 and GFAP) (Fig. EV2C). These results indicate that

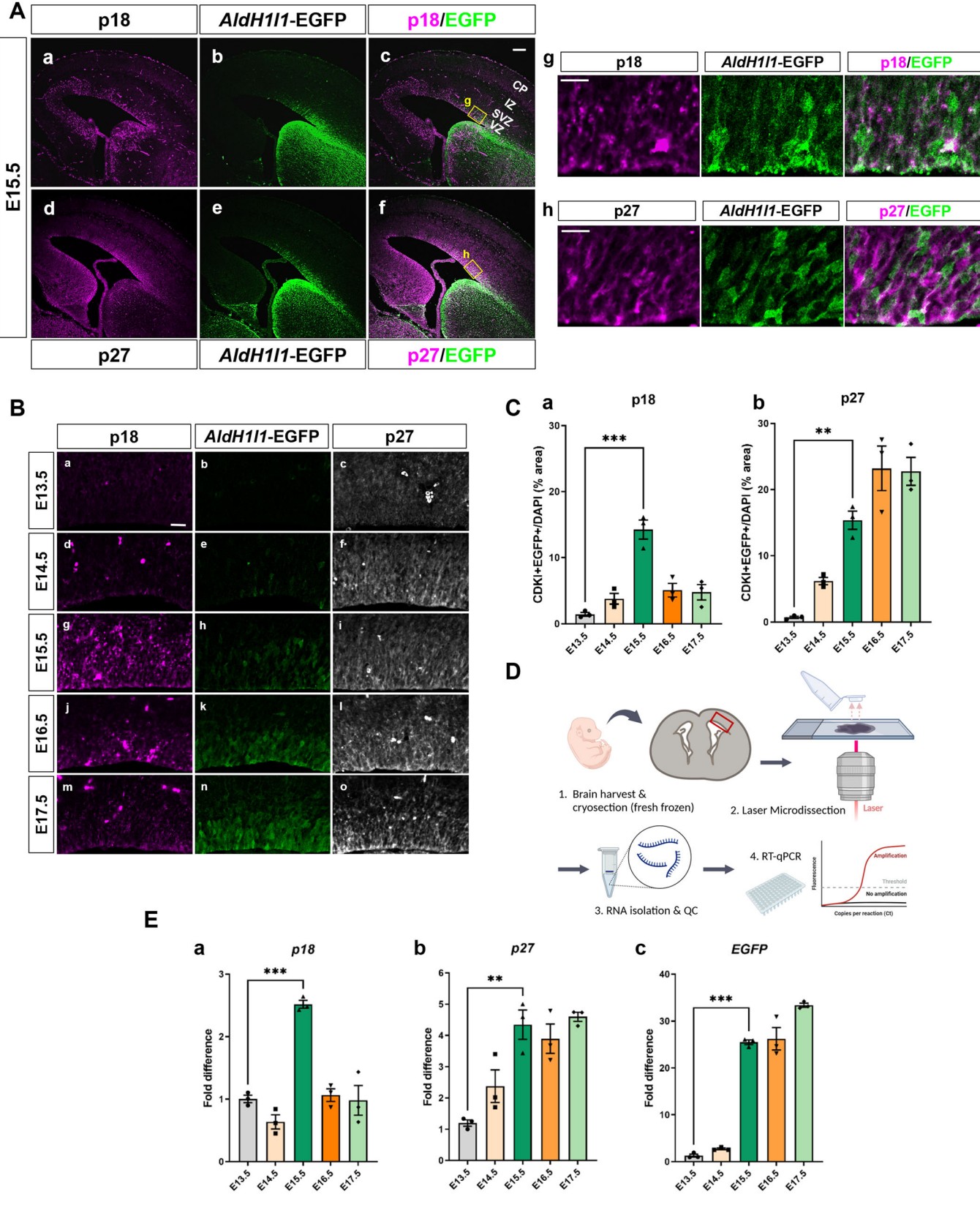

**Figure 1. Endogenous expression of p18 and p27 in the embryonic *AldH1l1*-EGFP mouse brain.**

(A) Low-magnification images of E15.5 brains of *AldH1l1*-EGFP mice stained with p18 (a, c) and p27 (d, f) antibodies. VZ, ventricular zone; SVZ, subventricular zone; IZ, intermediate zone; CP, cortical plate. Bar, 100 μm. High-magnification images of VZ stained with p18 (g) and p27 (h) antibodies. Bar, 20 μm. (B) Magnified view of (A) near the ventricular surface stained with p18 (a, d, g, j, m) and p27 (c, f, i, l, o) antibodies (E13.5–17.5). Bar, 30 μm. (C) Quantification of the CDKIs and EGFP double-positive area in (B) ($n = 3$ brains each, normalized by the DAPI-positive area). One-way ANOVA with Tukey's test; error bars show mean ± SEM. ***$P = 0.00003438$ (a), **$P = 0.00200336$ (b). (D) Schematics of laser microdissection of brain sections and RNA preparation (created with BioRender.com.). (E) qPCR analysis of p18, p27, and *AldH1l1*-EGFP expression in the VZ (E13.5-17.5, $n = 3$ brains). One-way ANOVA with Tukey's test; error bars show mean ± SEM. ***$P = 0.00007995$ (a), **$P = 0.00130536$ (b), ***$P = 0.00000024$ (c). Source data are available online for this figure.

p18 can induce astrocytes, which exhibit distinct migration patterns. Notably, a significant increase in Sox10- and PDGFRalpha-positive cells was not recognized, indicating that p18 OE did not induce the oligodendrocyte lineage (Fig. EV2D). We further tested whether p18 affects astrocyte differentiation in a cell-autonomous manner using an in vitro system of NS cells (Imayoshi et al, 2013). NS cells derived from *AldH1l1*-EGFP Tg mice were maintained in proliferative medium and then switched to astrocytic differentiation conditions after introducing p18-P2A-mKusabiraOrange2 (mKO2) or its control. The presence of p18 caused a significant increase in *AldH1l1*-EGFP expression, suggesting that p18 can induce astrocytic differentiation in a cell-intrinsic manner (Fig. 2H,I).

## The function of CDKIs in neural and astrocyte differentiation is stage-dependent

Our results indicate that p18 in apical progenitors at particular developmental stages upregulates *AldH1l1*-EGFP, an astrocyte marker. However, previous reports demonstrated that the introduction of CDKIs or lengthening of the cell cycle enhanced neural production (Calegari and Huttner, 2003; Dehay and Kennedy, 2007; Nguyen et al, 2006). To explore the seemingly conflicting results, we introduced CDKIs to apical progenitors in various stages during cortical development (Fig. 3A,B). When p18 and p27 were electroporated to E12.5, the number of cells positive for Neurogenin2 (Ngn2), a key transcription factor that generates excitatory neurons, increased significantly at E13.5 compared to the control. Notably, Ngn2-positive cells decreased at E15.5 and 16.5 after IUE at E14.5 and 15.5, respectively. This finding suggests that the OE of CDKIs at earlier stages can enhance neural lineage-committed cells while suppressing them at later stages.

There is a possibility that an enhanced signal of *AldH1l1*-EGFP by p18 OE could be a temporal effect and might not mean an increase in terminally differentiated astrocytes. We therefore electroporated p18 in embryonic stages and then dissected brains at P10 when cortical layers are virtually formed (Fig. 3C). We did not find significant changes in the cellular distributions and lineages between p18 OE and the control introduced at E12.5, the period when Ctip2-positive deeper-layer neurons are generated (Fig. 3D; Appendix Fig. S2). The control electroporated cells at E15.5 differentiated into upper-layer (UL) neurons, showing a distribution close to the pial surface due to radial migration. We found ectopic distributions of p18 OE cells at the ventricular surface and internal cortical region in addition to the upper layer. We further investigated the correlation between the position and fate of electroporated cells by using markers of UL neurons (Satb2 and Cux1) and astrocytes (GFAP and S100Beta). Satb2- and Cux1-positive cells in the upper layer were decreased, while GFAP- and

S100Beta-positive cells close to the ventricular surface were increased by p18 OE (Fig. 3E). These results suggest that the differentiation lineage was switched from the production of UL neurons to astrocytes by p18 OE introduced at E15.5.

We next examined morphological characteristics at single-cell resolution in the postnatal brain. A limited number of electroporated cells were designed to express membrane-localizing EGFP by Cre-loxP recombination (Fig. 4A). Subsequently, we performed program-based quantifications of the nuclear volume and aspects of the cellular process, including number, length, and directionality (Fig. 4B). When IUE was performed at E14.5 and 15.5 when UL neurons are generated, the nuclear volume was decreased by p18 OE. Importantly, control cells electroporated at E17.5, when astrocytes are primarily produced, showed a smaller nuclear volume (Fig. 4C). Moreover, cellular processes became shorter, and their numbers were increased by p18 OE at E14.5 and 15.5 (Fig. 4D,E). A comparison of process directionality per area divided by angle showed that the polarized shape tended to be lost by p18 OE, which reflects the nonpolar shape of astrocytes (Fig. 4F). Thus, in addition to changes in marker expression, morphological alterations were evoked as a consequence of p18 OE in apical progenitors, which originally ought to produce UL neurons.

## Generation of conditional p18-P2A-mKO2 knock-in mice to analyze lineage transition

Our results obtained by IUE indicate that p18 OE induces switching from neural to astrocytic differentiation in a developing stage-dependent manner. To analyze the molecular basis of the lineage transition in a more robust system than IUE, we generated a conditional knock-in mouse containing a loxP-flanked stop sequence and p18-P2A-mKO2 in the ROSA26 locus (Fig. 5A). We first delivered the Cre gene by IUE to confirm the production of p18 and mKO2 (Appendix Fig. S3A). We crossed the p18-P2A-mKO2 mouse to the Nestin-CreER$^{T2}$ line (Imayoshi et al, 2006) to allow the expression of p18-P2A-mKO2 in apical progenitors upon tamoxifen administration. The mKO2 signal was observed after Cre-loxP recombination in the brains of NesCreER$^{T2}$/p18-P2A-mKO2 double-positive mice but not in controls (Appendix Fig. S3B–F). To examine the effect of p18 in NSCs by histological analysis, we injected EdU into pregnant mice to label the proliferating cell population prior to tamoxifen administration at E15.5. The EdU incorporation patterns were essentially the same between the double-positive and control littermates. We found an increase in AldH1l1-positive cells and a decrease in Ngn2-positive cells among EdU-positive cells at E16.5 (Fig. 5B,C). Moreover, Satb2-positive cells in the UL were decreased, while AldH1l1- and Sox9-positive cells close to the ventricular surface increased in the P10 brains (Fig. 5D,E), validating that the lineage transition from

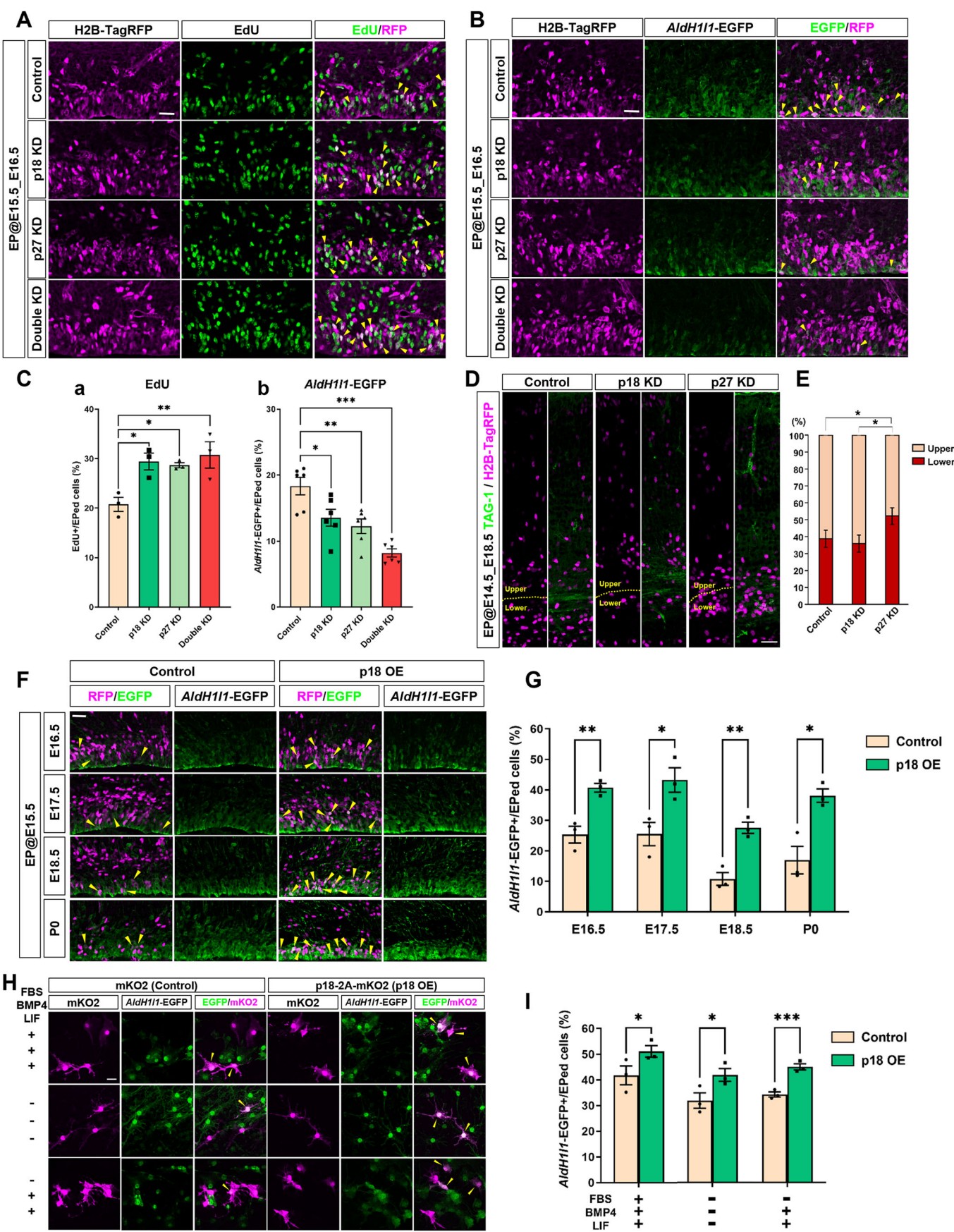

**Figure 2. Functional analysis of p18 and p27 in the developing dorsal cortex.**

(A) Plasmids to induce KD of p18 and p27 mixed with pCAG-H2B-TagRFP were delivered at E15.5 to *AldH1l1*-EGFP mice. EdU was administered 2 h before fixation at E16.5. Arrowheads; double-positive cells to EdU and H2B-TagRFP. Bar, 50 μm. (B) The expression of *AldH1l1*-EGFP prepared in (A). Arrowheads; double-positive cells to *AldH1l1*-EGFP and H2B-TagRFP. Bar, 50 μm. (C) Quantification of the stained images shown in (A) (*n* = 3 brains each) and (B) (*n* = 6 sections from 3 brains each). Total counted cells of Control, p18 KD, p27 KD, Double KD; *331, 433, 363, 420*. One-way ANOVA with Tukey's test; error bars show mean ± SEM. From left to right, *P = 0.02035068, *P = 0.03126960, **P = 0.00955672 (a), *P = 0.01734001, **P = 0.00268600, ***P = 0.00000817 (b). (D) Plasmids to induce KD of p18 and p27 mixed with pCAG-H2B-TagRFP were delivered at E14.5 and then fixed at E18.5. Sections were stained using TAG-1 antibody to recognize axon fibers as the border between the upper and lower positions. Bar, 50 μm. (E) Quantification of the stained images shown in (D) (*n* = 4 brains each). Total counted cells of Control, p18 KD, p27 KD; *544, 614, 429*. One-way ANOVA with Tukey's test; error bars show mean ± SEM. From left to right, *P = 0.04727532, *P = 0.01171081. (F) Plasmids to induce OE of p18 mixed with pCAG-H2B-TagRFP were delivered at E15.5 to *AldH1l1*-EGFP mice and then fixed at E16.5-P0. Arrowheads; double-positive cells to *AldH1l1*-EGFP and H2B-TagRFP. Bar, 50 μm. (G) Quantification of the stained images shown in (F) (*n* = 3 brains each). Total counted cells of Control, p18 OE; *256, 274* (E16.5); *400, 449* (E17.5); *390, 473* (E18.5); *253, 365* (P0). Two-tailed *t* test; error bars show mean ± SEM. From left to right, **P = 0.00752188, *P = 0.03272830, **P = 0.00392106, *P = 0.01368411. (H) pCAG-mKO2 or pCAG-p18-P2A-mKO2 were transfected into cultured NS cells prepared from *AldH1l1*-EGFP mouse brains, followed by induction to astrocytes using the conditions indicated on the left. Arrowheads; double-positive cells to *AldH1l1*-EGFP and H2B-TagRFP. Bar, 20 μm. (I) Quantification of the stained images shown in (H) (*n* = 3, independent culture each). Total counted cells of Control, p18 OE; *220, 296* (FBS + BMP4 + LIF + ); *140, 170* (FBS- BMP4-LIF-); *195, 380* (FBS-BMP4 + LIF + ). Two-tailed *t* test; error bars show mean ± SEM. From left to right, *P = 0.04486357, *P = 0.02332944, *P = 0.00392106, ***P = 0.00050115. Source data are available online for this figure.

UL neurons to astrocytes occurred in the p18 knock-in mouse brain, equivalent to the results of IUE.

## Global gene expression profiling in neural and glial differentiation by p18 in NSCs

To reveal the molecular and genetic network involved in neural and glial differentiation invoked by p18 OE, we designed a transcriptome analysis to compare differential gene expression in NSCs at earlier and later embryonic stages. We applied tamoxifen at E12.5 and 15.5 and then prepared RNA from NesCreER$^{T2}$/p18-P2A-mKO2 littermates (hereafter mKO2- or mKO2 + ) from the VZ in the dorsal cortex cut by LMD at E13.5 and 16.5, respectively (datasets were named *E13* and *E16*). The short induction period after tamoxifen administration enables the acquisition of gene sets expressed by the immediate response and minimizes the possibility of the inclusion of cells produced outside of the dorsal cortical VZ. A total of 2,828 and 354 genes were differentially expressed between *E16* and *E13* for mKO2- (equivalent to normal development) and mKO2 + , respectively (Fig. 6A,B), although comparisons between mKO2- and mKO2+ at each stage showed fewer differences (Fig. EV3A). As landmark molecules, we observed elevated expressions of neural markers (*Map2* and *Nptx1*) and astrocyte markers (*Aldh1l1* and *Gfap*) in *E13* and *E16*, respectively, in mKO2+ (Fig. EV3B). The expression of *p27* (*Cdkn1b*) showed no difference between mKO2- and mKO2+ (Fig. EV3C). Functional enrichment analysis (Fig. 6C,D) revealed that differentially expressed genes (DEGs) at *E16* relative to *E13* in mKO2- embryos (hereafter mKO2-$^{E16/E13}$) were enriched in biological pathway terms associated with the cell cycle (top 1st, 7th, and 11th) but not in mKO2 + $^{E16/E13}$, reflecting that the proliferation of NSCs in mKO2+ embryos was arrested. Importantly, notable pathways like "*Glial cell development*" (GO:0021782) and "*Delta-Notch signaling pathway*" (WP265) were identified in mKO2 + $^{E16/E13}$. Other pathways were largely overlapped in both mKO2-$^{E16/E13}$ and mKO2 + $^{E16/E13}$. In mKO2 + $^{E16/E13}$, DEGs involved in *Glial cell development*, such as *Adora2a* and *GFAP*, were upregulated, while *Dll1* and *Hes5*, related to the maintenance of stem cell features in the Notch signaling pathway, were downregulated, implying that differentiation to the glial lineage was promoted in mKO2 + $^{E16/E13}$.

To further compare the cell type composition, deconvolution of gene expression (BayesPrism) was performed combined with published mouse forebrain scRNA-seq data (La Manno et al, 2021) (Fig. EV3D). As a reference, clusters from the scRNA-seq data were annotated to distinct subtypes of Neuron (excitatory), Intermediate progenitor (Basal progenitor, positive for *Eomes/Tbr2*), Radial glia, and GABAergic (interneuron) by their marker genes (Fig. EV3E,F). A significant increase in the proportion of Neuron and a steep decrease in the proportion of Radial glia in mKO2+ compared to mKO2- were observed in *E13*, while there was no change in the proportion of Intermediate progenitor (Fig. 6E). This suggests that p18 promotes differentiation to the lineage of excitatory neurons at E13.5 by consuming apical progenitors. Notably, the proportion of Astrocyte was significantly increased in both *E13* and *E16*, while the proportions of Neuron, Radial glia, Intermediate progenitor, and Oligodendrocyte were neither increased nor decreased in *E16*. The proportion of GABAergic was also slightly increased in both *E13* and *E16*, although there was no statistical significance.

The top 50 differentially expressed genes were investigated to screen whether there were any functional differences in each cell type (Figs. 6F and EV3G). In Neuron, genes related to neural development, such as *Mef2c* and *Rbfox1*, were upregulated in mKO2+ only in *E13*, which might explain the experimental results that p18 OE induces the neural lineage at E13.5 but not at E16.5. In Astrocyte, upregulated expression of genes related to astrocyte development (*AldH1l1* and *Tnc*) was recognized in *E13*. Taken together with the results in Fig. 6E, it can be speculated that p18 OE may initiate astrocyte differentiation at E13.5. Of note, the proportion of increased Astrocyte was minor compared to the increased proportion of Neuron in *E13*, while only Astrocyte was increased in *E16* (Fig. 6E). Interestingly, Dlx-family genes, as well as marker genes of interneurons (*Gad1*, *Gad2*), were highly expressed in GABAergic in mKO2+ (Fig. 6F). These results demonstrate that p18 OE has a wide range of effects on the transcriptome related to the lineage transition of neurons and glial cells.

Using NesCreER$^{T2}$/p18-P2A-mKO2 mice, we also investigated whether the methylation status of genomic DNA changes by p18 OE. Although we found a decrease in CpG methylation at the promoter region of *GFAP* along the developmental stage as previously reported (Takizawa et al, 2001), we could not find significant differences between mKO2- and mKO2+ both at E13.5 and E16.5 (Fig. EV4).

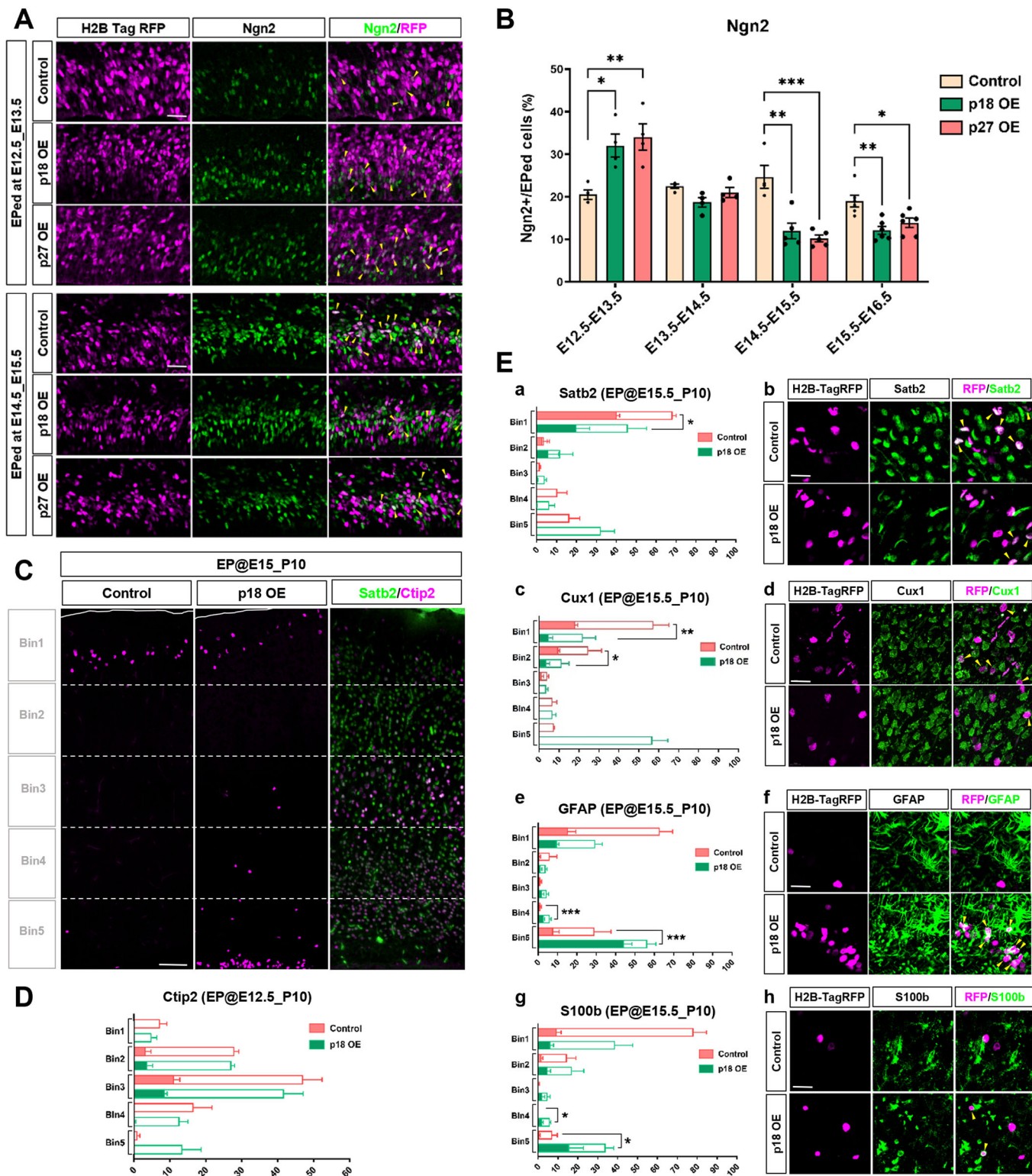

## Expression and function of Dlx2 in the dorsal cortical lineage

Our transcriptome analysis revealed that Dlx-family genes (*Dlx1, 2, 5,* and *6*) were upregulated at E16.5 in NesCreER[T2]/p18-P2A-mKO2 mice (Fig. 6F). We validated the upregulation of *Dlx2* in p18-

electroporated dorsal cortical cells by qPCR (Appendix Fig. S4A). Dlx-family genes have been well studied as neurogenic genes in the GE-originated cells (Achim et al, 2014). However, lineage analyses using transgenic reporter mice (Kohwi et al, 2007) and scRNAseq (Li et al, 2021) have shown that Dlx-family genes are also expressed in a certain number of cells in the dorsal cortical lineage. Notably,

◄ **Figure 3. Stage-dependent functions of CDKIs.**

(**A**) Plasmids to induce OE of p18 and p27 mixed with pCAG-H2B-TagRFP were delivered at E12.5, 13.5, 14.5, and 15.5 and then fixed after 1 d. Sections were stained using Ngn2 antibody. Images after IUE at E12.5 and 14.5 are presented. Arrowheads; double-positive cells to Ngn2 and H2B-TagRFP. Bar, 50 μm. (**B**) Quantification of Ngn2-positive cells among H2B-TagRFP-positive cells ($n = 4$–6 brains). Total counted cells of Control, p18 OE, p27 OE; *262, 257, 155* (E12.5EP); *262, 199, 269* (E13.5EP); *229, 274, 225* (E14.5EP); *443, 375, 450* (E15.5EP). One-way ANOVA with Tukey's test; error bars show mean ± SEM. From left to right, *$P = 0.02269181$, **$P = 0.00939859$, **$P = 0.00130391$, ***$P = 0.00046411$, **$P = 0.00199138$, *$P = 0.01730243$. (**C**) Plasmids to induce OE of p18 with pCAG-H2B-TagRFP were delivered at E15.5 and then fixed at P10. Bar, 100 μm. (**D**) Quantification of Ctip2-positive cells among H2B-TagRFP-positive cells. Plasmids to induce OE of p18 with pCAG-H2B-TagRFP were delivered at E12.5 and then fixed at P10 ($n = 3$ brains). Total counted cells of Control, p18 OE; *975, 1282*. Two-tailed $t$ test; error bars show mean ± SEM. No statistically significant differences were detected. (**E**) (a, c, e, g) Quantification of Satb2 (a), Cux1 (c), GFAP (e), and S100beta (g)-positive cells among H2B-TagRFP-positive cells ($n = 3$–7 brains). Total counted cells of Control, p18 OE; *381, 639* (Satb2); *195, 194* (Cux1); *281, 560* (GFAP); *409, 488* (S100beta). Two-tailed $t$ test; error bars show mean ± SEM. From top to bottom, *$P = 0.04268962$ (a), **$P = 0.00460726$, *$P = 0.02607529$ (c), ***$P = 0.00056828$, ***$P = 0.00001354$ (e), *$P = 0.01101942$, *$P = 0.01365471$ (g). (b, d, f, h) Magnified images of sections stained by Satb2 (b; BIN1), Cux1 (d; BIN1), GFAP (f; BIN5), and S100beta (h; BIN5). Bar, 20 μm. Arrowheads; double-positive cells to H2B-TagRFP and Satb2 (b), Cux1 (d), GFAP (f), or S100b (h). (**C–E**) the entire span of the cortex was divided equally into 5 portions (BIN). BIN 1: pial side/BIN 5: ventricular side (same for Figs. 5D,E and 8C). The open (outside) bars indicate the ratio of H2B-TagRFP-positive cells in each BIN among all BINs, whereas the solid (inside) bars indicate positive cells (Ctip2 for **D**; Satb2, Cux1, GFAP, and S100beta for **E**) among RFP-positive cells. $t$ tests were performed for solid bars in the same BIN. Source data are available online for this figure.

the latter analysis reported that Dlx-family genes are expressed in multipotent intermediate progenitor cells (MIPCs), which can produce OB interneurons.

Here, we addressed the expression pattern of endogenous Dlx2 using Emx1-Cre/Ai9 mice, which labels dorsal cortex-originated cells with tdTomato (Fig. 7A). Interestingly, in the VZ and the subventricular zone (SVZ) at E17.5, 35% or 41% of Dlx2- or Ngn2-positive cells, respectively, were tdTomato positive, but only a few cells (<6%) were triple positive for Ngn2, Dlx2, and tdTomato (Fig. 7A). We then performed IUE targeting the dorsal VZ at E15.5. Brains dissected at E16.5-18.5 showed that, in controls, 5-7% of EGFP-positive cells were Dlx2-positive cells; those were increased to 14% at the perinatal stage in the case of p18 OE (Fig. 7B,C). We next conducted functional tests of Dlx2 in dorsal cortex-derived cells using Dlx2 OE and KD (Appendix Fig. S4B) constructs. We found that the number of Ngn2-positive cells was decreased by Dlx2 OE while increased by Dlx2 KD (Fig. 7D,E). We further explored whether Dlx2 can directly affect the promoter of Ngn2 by luciferase assay. Indeed, Dlx2 suppressed the promoter activity of Ngn2 as Hes1, a well-characterized suppressor of Ngn2 (Shimojo et al, 2008), did (Fig. 7F). Notably, the number of AldH1l1-positive cells was decreased by Dlx2 OE while increased by Dlx2 KD (Fig. 7G,H).

## Sequential regulation of lineages by p18: UL neurons, astrocytes, and OB interneurons

Functional analysis of Dlx2 raised the possibility that augmented Dlx2 by p18 might subsequently induce OB interneurons by suppressing astrocytes in the later stage. To characterize the mutual relations of each lineage, we performed OE of p18 and Dlx2 at E15.5 followed by immunostaining of E18.5 brains using markers of UL neurons (Satb2), astrocytes (AldH1l1 and Sox9), and OB interneurons (Sp8) (Fig. 8A,B). Both p18 and Dlx2 OE suppressed UL neurons; astrocytes were increased by p18 and decreased by Dlx2. OB interneurons were strongly increased by Dlx2. Interestingly, more cells were located in the basal region of the cortex in the control group and close to the ventricular surface in the p18 OE group and in the SVZ in the Dlx2 OE group (Fig. 8A), indicating that the induction of different lineages altered cellular localization. We further addressed the effects of p18 and Dlx2 OE in the postnatal stages. Upregulation of Dlx2 by p18 OE was prominent near the ventricular surface at P1 (Fig. 8C). Importantly, p18 and Dlx2 OE cells were identified in the rostral migratory stream (RMS) with Sp8 expression at P4, displaying their ability to migrate toward the OB (Fig. 8D).

The results of immunostaining and qPCR revealed that p18/p27 double KD increased UL neurons and reduced OB interneurons (Fig. 8E,F; Appendix Fig. S5). We next investigated the effect of Dlx2 KD on lineage determination. In contrast to Dlx2 OE, Dlx2 single KD increased UL neurons and astrocytes compared to the control at E18.5 (Fig. 8G,H). Since p18 enhanced the expression of Dlx2, we addressed the possibility of whether the removal of Dlx2 can strengthen the effect of p18 OE to elevate astrocyte induction. We performed combined IUE of p18 OE and Dlx2 KD, which resulted in more astrocytes than p18 single OE (Fig. 8G,H).

Finally, to further clarify whether the lineage transitions occur at the single cell level or distinct subsets in each lineage, we utilized piggyBac (PB)-*Gfa2*-EGFP, the transposon-based fluorescent reporter of the astrocyte lineage (Hamabe-Horiike et al, 2021). Chromosome-integrated PB-RFP introduced by IUE allows labeling all clones derived from electroporated NSCs. Once astrocyte lineage is induced, the cell becomes EGFP-positive, then the EGFP signal remains as the footprint. We verified that the *Gfa2*-EGFP was upregulated by p18 OE while downregulated by p18/p27 double KD, similar to AldH1l1 and Sox9 (Fig. 9A–C). Remarkably, Dlx2/RFP double-positive cells with *Gfa2*-EGFP signal were prominently increased by p18 OE at the perinatal stage (E17.5 and P1) (Fig. 9D,E). Moreover, Sp8-positive cells in the RMS were increased by p18 OE (28% among RFP-positive cells) compared to the control (14%) at P4. With p18 OE, 67% of Sp8/RFP double-positive cells exhibited the EGFP signal (Fig. 9D,E). These results confirm the view that p18 facilitates lineage transitions from NSC to OB interneuron via astrocyte within a single cell's progeny.

## Discussion

### CDKIs, cell cycle, and differentiation

The conceptual novelty of this study can be summarized as follows: (1) the role of CDKIs differs between early and late stages of brain development; (2) p18 facilitates lineage transitions of excitatory neurons, astrocytes, and interneurons. Regarding the first point, we

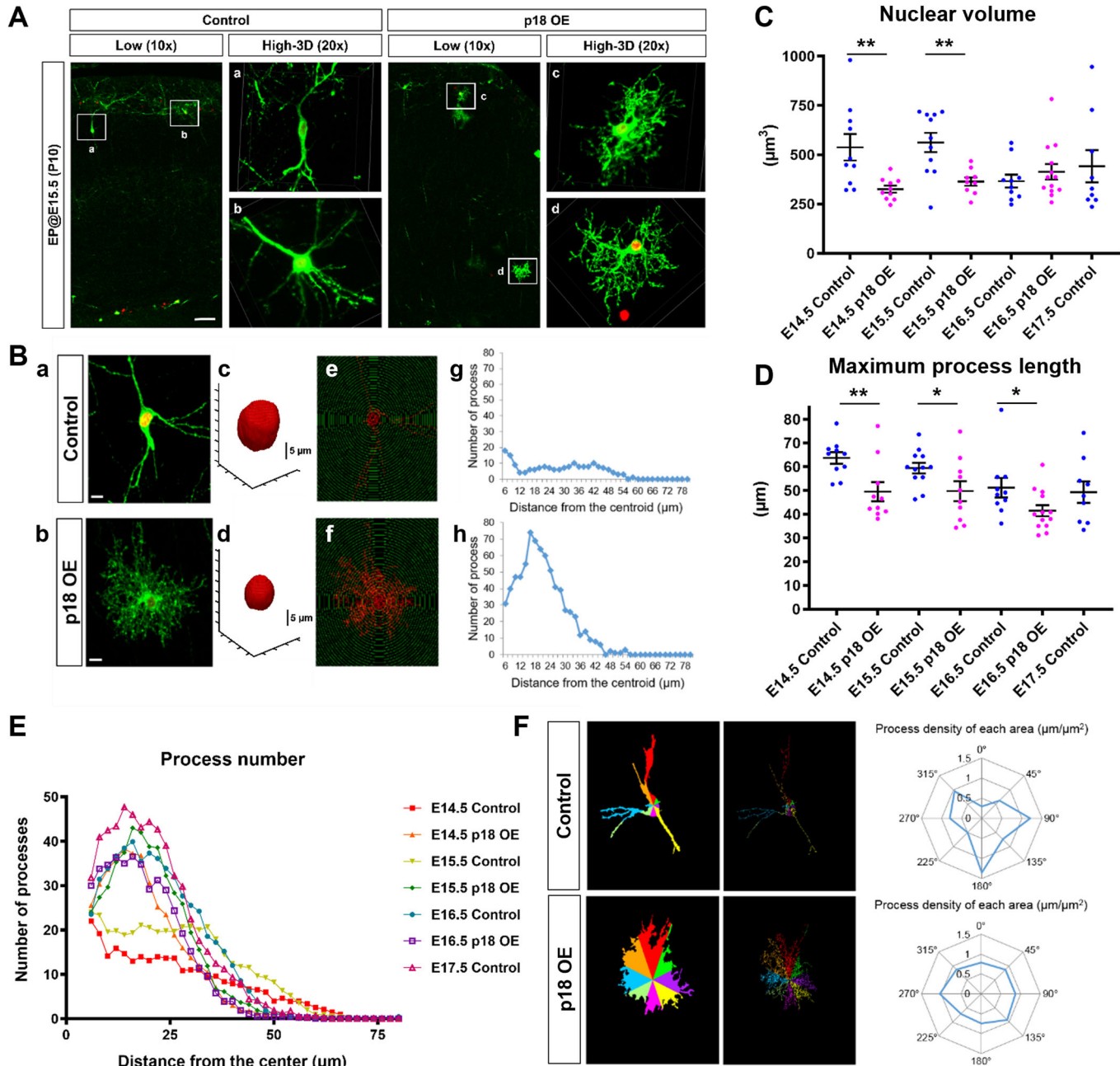

**Figure 4. Quantitative morphological analysis of p18 OE cells at single-cell resolution.**

(A) For the labeling of individual cell shapes, plasmids to induce OE of p18 with pCAG-H2B-TagRFP, pCAG-FloxP-EGFP-F, and pCAG-Cre were delivered at E14.5-17.5 and then fixed at P10. Low magnification and 3D-reconstructed images are exhibited. Bar, 50 μm. (B) Examples of the morphological analysis. Cells without overlapping were chosen (a and b), and then the nuclear shape was deconvoluted (c and d). After the calculation of the centroid, a Sholl analysis using the 2D-projected image was performed (e and f) and then analyzed (g and h). Bar, 10 μm. (C) Comparison of nuclear volume at P10 after IUE at E14.5, 15.5, and 16.5 for the control and p18 (n = 10, 10, 11, 9, 10, 13, 9, from the left of the graph). For the control, the IUE at E17.5, the stage of dominant production of endogenous astrocytes, was also tested. Two-tailed t test; error bars show mean ± SEM. From left to right, **P = 0.00666581, **P = 0.00293175. (D) Comparison of maximum process length among the conditions indicated in (C) (n = 10, 10, 12, 10, 10, 13, 9, from the left of the graph). Two-tailed t test; error bars show mean ± SEM. From left to right, **P = 0.00738915, *P = 0.04504341, *P = 0.04259063. (E) Results of the Sholl analysis. Averaged numbers of 9 samples for each condition are presented. (F) Examples showing the relation between the angle and density of processes (scored by the total length of processes per colored area) at P10 after the IUE at E15.5. Source data are available online for this figure.

found that p18 induced neurons in the earlier stage of cortical development and astrocytes in the later stage with reduced UL neurons, while loss of p18/p27 function had the opposite effect. Regarding the second point, surprisingly, p18 evoked the expression of Dlx-family genes, which subsequently induced OB interneuron and suppressed excitatory neurons and astrocytes at the perinatal stage. Thus, our study describes the integrated function of p18 to determine the lineage boundaries (excitatory

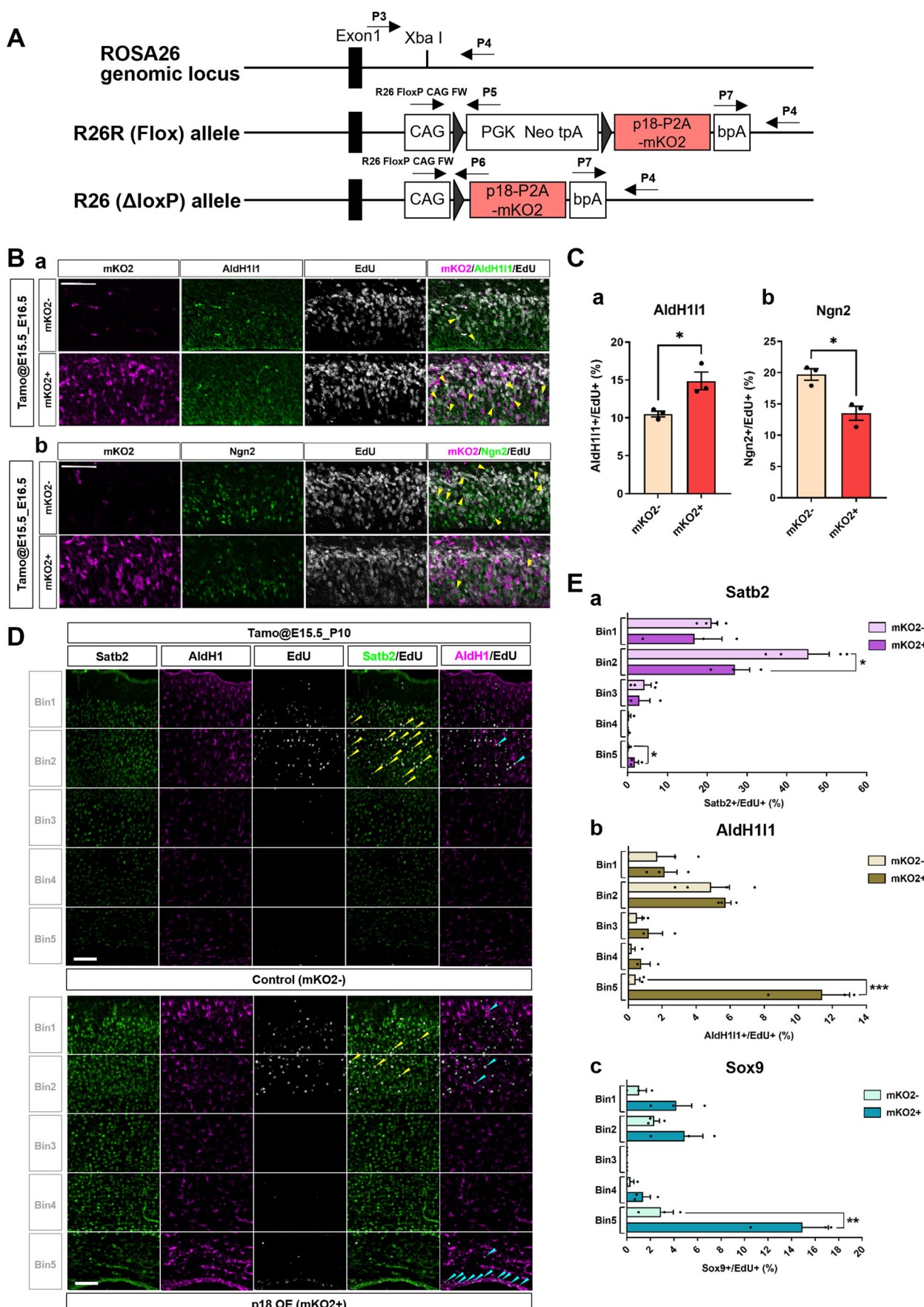

**Figure 5. Generation and characterization of R26R-p18-P2A-mKO2 Tg mice.**

(A) Constructs for Tg mouse generation. Arrows indicate primers for genotyping. (B) To label proliferating neural progenitors, EdU was injected into pregnant R26R-p18-P2A-mKO2/Nes-CreER[T2] mice 4 h prior to tamoxifen administration at E15.5, followed by fixation at E16.5. Sections were stained using AldH1l1 (a) and Ngn2 (b) antibodies with EdU detection. Arrowheads; double-positive cells to EdU and AldH1l1 (a) or Ngn2 (b). Bar, 100 μm. (C) Quantification of the stained images shown in (B) ($n = 3$ brains each). Total counted cells of mKO2-, mKO2 + ; 571, 523 (AldH1l1); 488, 444 (Ngn2). Two-tailed $t$ test; error bars show mean ± SEM. *$P = 0.02391804$ (a), *$P = 0.01248425$ (b). (D) Mice were prepared as indicated in (B), and then pups were fixed at P10. Sections were stained using Satb2, AldH1l1, and Sox9 antibodies with EdU detection. Arrowheads; double-positive cells to EdU and Satb2 (yellow) or AldH1l1 (light blue). Bar, 100 μm. (E) Quantification of images in (D). The bars indicate the proportions of Satb2 (a)-, AldH1l1 (b)-, and Sox9 (c)-positive cells among EdU-positive cells in each BIN ($n = 3-4$ brains each). Total counted cells of mKO2-, mKO2 + ; 385, 486 (Satb2 + /EdU + , AldH1l1 + /EdU + ); 305, 345 (Sox9 + /EdU + ). Two-tailed $t$ test; error bars show mean ± SEM. From top to bottom, *$P = 0.04270011$, *$P = 0.04262900$ (a), ***$P = 0.00024186$ (b), **$P = 0.00392151$ (c). Source data are available online for this figure.

neurons/glial cells/OB interneurons) that emerge sequentially in the dorsal cortex (Fig. 9F, model), which could fill one of the missing links in the lineage specification of multipotent NSCs. Of note, not all p18-activated cells eventually turn into OB interneurons by functions of Dlx-family genes. We consider that the following factors are likely to affect lineage determination between astrocytes and OB interneurons: (1) genetic and epigenetic elements of astrocyte differentiation (see Introduction) and (2) the timing of OB interneuron induction via Dlx-family genes.

Previous studies of CDKIs in regard to neural fate mostly investigated the Cip/Kip family rather than the Ink4 family. Here, we explored the functions of p18 (Ink4) and p27 (Cip/Kip), which show increased expression in NSCs at the onset of glial production. Both the Ink4 and Cip/Kip families regulate CyclinD-Cdk4/6, while Cip/Kip also inactivates CyclinE-Cdk2, CyclinA-Cdk2, and CyclinB-Cdk1 (Grison and Atanasoski, 2020; Schirripa et al, 2022). Then, what are the roles of components other than CDKIs in NSCs? Double Cdk4/Cdk6 KO mice showed a reduction in intermediate progenitors, while Cdk2/Cdk6 had little effect on progenitor proliferation (Grison et al, 2018). Double KD of Cdk4 and CyclinD1 caused a prolonged cell cycle of apical progenitors and increased neurogenesis (Lange et al, 2009). More recently, it was reported that CyclinB1/2 and CyclinD1 have significant roles in controlling the differentiation of cortical progenitors (Hagey et al, 2020). The expression of Slc1A3/Glast, a marker of RGCs and astrocytes (Mori et al, 2006), was downregulated by CyclinD1 gain-of-function while upregulated by its loss-of-function. This result is consistent with our findings that the acquisition of glial cell features is regulated by cell cycle controllers. CyclinB1/2 had opposite effects, which is explained by the altered cell division mode (symmetric vs. asymmetric) of NSCs and Notch signaling (Hagey et al, 2020). Our in vitro study (Fig. 2) supports the view that the p18-dependent regulation of differentiation can be autonomous. However, the combined effects of autonomous and non-autonomous might be involved in vivo. Indeed, our RNAseq analysis of microdissected tissue implies the involvement of the Delta-Notch pathway in p18 OE conditions (Fig. 6). Further comprehensive analyses would uncover the mutual relationships of CDKIs, cyclins, and other cell cycle regulators to control lineage transitions of cortical NSCs.

## Multilineage potency of NSCs in the dorsal cortex

Traditionally, it has been well described that excitatory neurons are produced in the dorsal cortex, while GABAergic interneurons are generated in the GE (Achim et al, 2014). In particular, OB

interneurons are generated at the LGE and reach the OB via the RMS (Waclaw et al, 2006). As mentioned in the Results, a more recent lineage study using scRNAseq revealed the existence of MIPCs in the dorsal cortical wall at the perinatal stage (Li et al, 2021; Zheng et al, 2022). At approximately E16.5, RGCs start to transform into apical MIPCs (aMIPCs). Subsequently, aMIPCs rapidly become bMIPCs, which can give rise to astrocytes, oligodendrocytes, and OB interneurons. Notably, the scRNA-seq results indicate that the expression of Mki67 and Cdk1, markers of cell proliferation, is rarely detectable in aMIPCs (Li et al, 2021). Given that p18 inhibits cell cycle progression, a reduction in proliferation might be required to transform RGCs into aMIPCs. Another interesting point is that bMIPCs gradually show the features of OB intermediate progenitor cells by expressing Dlx-family genes and then eventually differentiate into Sp8/9-positive OB interneurons (Li et al, 2021). Here, our results showed that Dlx2 OE suppressed the lineage of astrocytes, while loss of Dlx2 function induced more astrocyte populations. These findings suggest that the expression of Dlx-family genes in bMIPCs can determine the lineage between astrocytes and OB interneurons. It has also been reported that forced expression of Dlx2 at the earlier developmental stage induces local accumulation of cells showing OB interneuron markers (Guo et al, 2019). Importantly, our results demonstrate that OB interneurons produced by Dlx2 OE reach the OB through chain migration at the neonatal stage (Fig. 8). Taken together, it can be deduced that the expression of Dlx-family genes should be at the proper timing to produce interneurons with migration ability to reach the OB.

## Influences of lineage choice on brain size and architecture

Our results demonstrate that the expression level of p18 determines the boundary among lineages that appear from NSCs in the developing mammalian cortex. As a result, the ratio and localization of each cell type born at the given time differed in the late embryonic and postnatal stages. Based on the findings, we deduce a unique possibility that the timing of CDKI expression in the developing stage can be one of the vital factors that influence diverse brain sizes and architecture among species. Indeed, a previous study using KO mice described the increased size of general organs in p18 and p27 null mice, although a detailed histological analysis of the brain architecture is still required (Franklin et al, 1998). Investigations of spatiotemporal expression patterns of CDKIs in the developing brains of several species and manipulations of their functions in vivo using model animals or

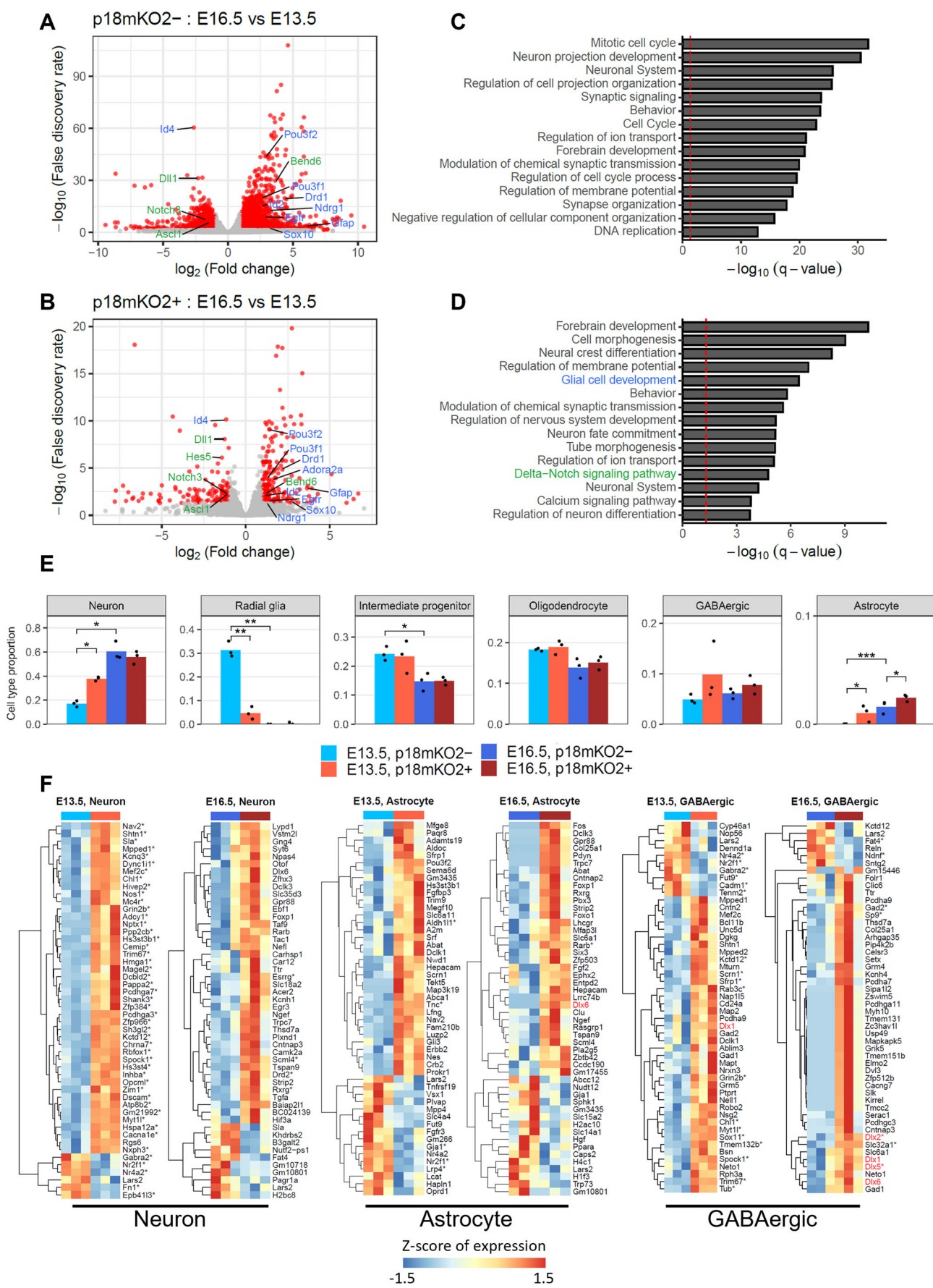

◄

**Figure 6.   Cell type proportion and transcriptomic change by p18 OE.**

(A, B) Volcano plots of RNA-seq for each comparison. Red dots ($n = 2828$ for **A** and $n = 354$ for **B**) indicate that the difference in gene expression is statistically significant (false discovery rate (FDR) < 0.05, |fold change (FC)| > 2). Genes that correspond to the biological pathways indicated in (**D**) are marked by identical colors. (**C, D**) Functional enrichment of differentially expressed genes. The Q-value represents an FDR. (**E**) Cell type proportion estimated by BayesPrism using mouse brain scRNA-seq data as the reference. The significance of the difference was calculated using the relative proportion to endothelial cells. One-tailed $t$ test. From top to bottom, left to right, *$P = 0.0276$, *$P = 0.0108$ (Neuron), **$P = 0.000270$, **$P = 0.000690$ (Radial glia), *$P = 0.0478$ (Intermediate progenitor), ***$P = 0.0000341$, *$P = 0.00471$, *$P = 0.0280$ (Astrocyte). (**F**) Heatmaps of the gene expression z score from the 50 most variable genes between p18mKO2- and p18mKO2 +. Asterisks indicate that the difference in expression is statistically significant (FDR < 0.05). Source data are available online for this figure.

in vitro organoid models are considerable strategies to address this possibility.

Alternatively, comparative analyses of dissociated single-cell transcriptomes among species would be an informative approach. Recently, scRNAseq profiles focusing on the glial production stage of the developing human brain by combining published datasets and immunohistochemistry have been reported (Yang et al, 2022). In their study, the existence of bMIPCs, which can give rise to astrocytes, oligodendrocytes, and OB interneurons, was confirmed in human cells. In the case of mouse brain development, bMIPCs are derived from RGCs via aMIPCs. In humans, RGCs existing in the VZ (vRGs) produce outer RGCs (oRGs) and truncated RGCs (tRGs). oRGs exclusively produce neurons, while tRGs generate neurons, glial cells, and bMIPCs. Notably, oRGs are more proliferative than tRGs (Nowakowski et al, 2016), indicating that the cell cycle of tRGs is longer than that of oRGs. Therefore, the diversity of progenies from tRGs may correlate to the functional level of CDKIs to determine the different constitutions of neurons and glial cells among species. Further studies of the cell cycle and its regulatory machinery will shed light on the evolutionary differences in brain architectures.

### Limitations of the study

In this study, we demonstrated the function of p18 in determining the boundaries among different cellular lineages sequentially arising from NSCs. However, further comprehensive understandings of the dynamics of endogenous CDKIs along the lineage transition require monitoring not only gene expression directly (possibly using live imaging by establishing mouse lines with stably expressing fluorescent reporter) but also proteostasis, as CDKIs are subject to ubiquitination during the cell cycle (Forget et al, 2008).

## Methods

### Reagents and tools table

| Reagent/resource | Reference or source | Identifier or catalog number |
| --- | --- | --- |
| **Experimental models** | | |
| Mouse: *AldH1l1*-EGFP | Mutant Mouse Resource & Research Centers (MMRRC) | Cat #011015-UCD |
| Mouse: P18-P2A-mKO2 | This study (RIKEN Center for Biosystems Dynamics Research) | Accession No. CDB1348K |
| Mouse: Nestin-CreER^T2 | Kyoto University | N/A |

| Reagent/resource | Reference or source | Identifier or catalog number |
| --- | --- | --- |
| Mouse: Ai9tdTomato | Korea Brain Research Institute | N/A |
| Mouse: Emx1-Cre | Jackson Laboratory | Cat #005628 |
| Mouse: ICR | Orient Bio/Core Tech | |
| Mouse: Neural Stem Cell | Imayoshi et al, 2013 | N/A |
| Human: HEK293T Cell | ATCC | CRL-3216 |
| **Recombinant DNA** | | |
| pCAGGS | Niwa et al, 1991 | N/A |
| pCAG-NLS-EGFP | Konno et al, 2008 | N/A |
| pCAG-p18 | Kosodo et al, 2011 | N/A |
| pCAG-FloxP-EGFP-F | Shitamukai et al, 2011 | N/A |
| pCAG-Cre | Shitamukai et al, 2011 | N/A |
| LacZmiRNA | Kosodo et al, 2011 | N/A |
| HA-HES1 | Shimojo et al, 2008/ Kobayashi et al, 2015 | N/A |
| E1-pNgn2-luc-3'UTR | Shimojo et al, 2008/ Kobayashi et al, 2015 | N/A |
| PB-*Gfa2*-EGFP | Hamabe-Horiike et al, 2021 | N/A |
| pCAG-p18 (human) | This paper | N/A |
| pCAG-H2B-TagRFP | This paper | N/A |
| pCAG-p18-P2A-mKO2 | This paper | N/A |
| pCAG-p27 | This paper | N/A |
| pCAG-CreER^T2 | Matsuda and Cepko, 2007 | N/A |
| pCAG-Dlx2 | This paper | N/A |
| **Antibodies** | | |
| Rabbit anti-AldH1l1 (1:500) | Abcam #ab87117 | AB_10712968 |
| Goat anti-TAG-1 (1:50) | DSHB #4D7/TAG1 | AB_531775 |
| Rabbit anti-Neurogenin2 (1:200) | CST #13144 | AB_2798130 |
| Rabbit anti-GFAP (1:500) | Dako #Z0334 | AB_10013382 |
| Mouse anti-S100b (1:500) | Abcam #ab11178 | AB_297817 |
| Mouse anti-Satb2 (1:500) | Abcam #ab51502 | AB_882455 |
| Rabbit anti-Cux1 (1:500) | Proteintech #11733-1-AP | AB_2086995 |
| Rat anti-Ctip2 (1:200) | Abcam #ab18465 | AB_2064130 |
| Rat anti-PECAM (1:500) | BD Pharmingen #550274 | AB_393571 |
| Rat anti-PDGFRa (1:500) | Invitrogen #16-1401-82 | AB_529483 |
| Rabbit anti-SOX10 (1:500) | CST #89356 | AB_2792980 |

| Reagent/resource | Reference or source | Identifier or catalog number |
|---|---|---|
| Mouse anti-Dlx2 (1:500) | Santa Cruz #sc-393879 | |
| Rabbit anti-Sox9 (1:500) | Millipore #AB5535 | AB_2239761 |
| Rabbit anti-Sp8 (1:500) | Novus #NBP2-49109 | |
| Rabbit anti-p18$^{Ink4c}$ (1:1000) | Santa Cruz #sc-865 | AB_2078731 |
| Rabbit anti-p27$^{Kip1}$ (1:1000) | Invitrogen #PA-5-27188 | AB_2544664 |
| Donkey anti-Chicken IgG, Alexa Fluor 488 conjugated (1:1000) | Jackson Immuno Research #703-545-155 | AB_2340375 |
| Donkey anti-rat IgG, Alexa Fluor 488 conjugated (1:1000) | Invitrogen #A21208 | AB_141709 |
| Donkey anti-rabbit IgG, Alexa Fluor 488 conjugated (1:1000) | Invitrogen #A21206 | AB_2535792 |
| Donkey anti-mouse IgG, Alexa Fluor 488 conjugated (1:1000) | Invitrogen #A21202 | AB_141607 |
| Donkey anti-rat IgG, Alexa Fluor 555 conjugated (1:1000) | Invitrogen #A48270 | AB_2896336 |
| Donkey anti-goat IgG, Alexa Fluor 555 conjugated (1:1000) | Invitrogen #A21432 | AB_2535853 |
| Donkey anti-rabbit IgG, Alexa Fluor 555 conjugated (1:1000) | Invitrogen #A31572 | AB_162543 |
| Donkey anti-mouse IgG, Alexa Fluor 555 conjugated (1:1000) | Invitrogen #A31570 | AB_2536180 |
| Donkey anti-rat IgG, Alexa Fluor 647 conjugated (1:1000) | Invitrogen #A78947 | AB_2910635 |
| Donkey anti-rabbit IgG, Alexa Fluor 647 conjugated (1:1000) | Invitrogen #A31573 | AB_2536183 |
| Donkey anti-mouse IgG, Alexa Fluor 647 conjugated (1:1000) | Invitrogen #A31571 | AB_2162542 |
| **Oligonucleotides and other sequence-based reagents** | | |
| Primers | This paper | N/A |
| **Chemicals, enzymes and other reagents** | | |
| 5-ethynyl-2'-deoxyuridine (EdU) | Invitrogen | Cat #A10044 |
| Corn oil | Sigma | Cat #C8267 |
| 4% Paraformaldehyde | Chembio | Cat #CBPF-9004 |
| Sucrose | Kisanbio | Cat #MB-S4842-1 |
| Bovine Serum Albumin | Sigma | Cat #A9647 |
| Triton X-100 | Sigma | Cat #X100 |
| FastStart Universal SYBR Green Master | Roche | Cat #0913850001 |
| Papain | Worthington | Cat #LK003153 |
| Poly-L ornithine | Sigma | Cat #P3655 |

| Reagent/resource | Reference or source | Identifier or catalog number |
|---|---|---|
| Fibronectin | R&D Systems | Cat #1030-FN-05M |
| ViaFect transfection reagent | Promega | Cat #E4982 |
| DMEM/F12 | Gibco | Cat #11039-021 |
| Penicillin/streptomycin | Gibco | Cat #15140-122 |
| N2 plus media supplement | R&D Systems | Cat #AR009 |
| Fibroblast Growth Factor (basic) (bFGF) | Wako | Cat #060-04543 |
| Epidermal Growth Factor (EGF) | Invitrogen | Cat #53003-018 |
| B-27 Supplement Minus Vitamin A | Invitrogen | Cat #12587-010 |
| Leukemia Inhibitory Factor (LIF) | Millipore | Cat #ESG1106 |
| Recombinant Human BMP-4 Protein | R&D Systems | Cat #314-BP-010 |
| Fetal Bovine Serum (FBS) | Gibco | Cat #16000-044 |
| Entobar (Sodium Pentobarbital) | Hanlim Pharm | |
| DMEM with high glucose | Welgene | Cat #LM001-09 |
| PEI-MAX | Polysciences | Cat #24765-1 |
| Opti-MEM | Invitrogen | Cat #31985-062 |
| **Software** | | |
| NIS-Elements | Nikon Instruments Inc | N/A |
| Imaris | Oxford Instruments | N/A |
| Adobe Photoshop | Adobe Systems | N/A |
| Metamorph | Molecular Devices | N/A |
| ImageJ | ImageJ | https://imagej.nih/gov/ij |
| GraphPad Prism 10 | GraphPad | https://www.graphpad.com |
| MATLAB | MathWorks | N/A |
| Kaluza | Beckman Coulter | N/A |
| RSEM/STAR | Incodom | https://incodom.kr |
| DESeq2 | Bioconductor | https://bioconductor.org |
| Metascape | Metascape | https://metascape.org |
| BayesPrism | BayesPrism | https://bayesprism.org |
| BioRender | BioRender | https://www.biorender.com |
| Adobe Illustrator | Adobe Systems | N/A |
| **Other** | | |
| Tyramide Signal Amplification Kit | Molecular Probes | Cat #40993 |
| Click-iT EdU imaging Kit | Invitrogen | Cat #C10340 |
| NucleSpin RNA XS | Macherey-Nagel | Cat #740902.50 |
| PureLink Hipure Plasmid Maxiprep Kit | Invitrogen | Cat #K2100-07 |

| Reagent/resource | Reference or source | Identifier or catalog number |
|---|---|---|
| ReverTra Ace qPCR RT Kit | TOYOBO | Cat #FSQ-201 |
| Dual-Luciferase Reporter Assay System | Promega | Cat #E1910 |
| MethylEasy™ Xceed Rapid DNA Bisulfite Modification Kit | Genetic Signatures | |
| NETflex Rapid Directional RNA-Seq Kit | PerkinElmer | Cat #NOVA-5198-10 |

## Methods and protocols

### Animals

The *AldH1l1*-EGFP line was imported from Mutant Mouse Resource & Research Centers (MMRRC, USA); the p18-P2A-mKO2 knock-in (KI) mouse line was generated and imported from RIKEN Center for Biosystems Dynamics Research (Japan); the Nestin-CreER$^{T2}$ line (Imayoshi et al, 2006) was kindly provided by Dr. Ryo-ichiro Kageyama (Kyoto Univ, Japan); the Ai9tdTomato line was housed at the KBRI; and the Emx1-Cre line was purchased from the Jackson Laboratory (#005628, USA). Pregnant ICR mice were purchased from Orient Bio or Core Tech (Republic of Korea). The plug date was considered embryonic day (E) 0.5. For the EdU assay, EdU (A10044, Invitrogen) was dissolved in PBS and administered by intraperitoneal (IP) injection at a concentration of 5 mg/kg. Tamoxifen (T5648, Sigma) was dissolved in corn oil (C8267, Sigma) at a concentration of 10 mg/ml, and 2 mg was injected into pregnant mice by IP injection. All experiments were approved by the animal care and use committee of KBRI (IACUC-22-00027) and the Institutional Animal Care and Use Committee of RIKEN Kobe Branch (A2001-03).

### Generation of p18-P2A-mKO2 knock-in mice

p18-P2A-mKO2 mice (Accession No. CDB1348K: https://large.riken.jp/distribution/mutant-list.html) were generated by using HK3i embryonic stem (ES) cells derived from C57BL/6 (Kiyonari et al, 2010). The targeting vector was generated with the Gateway system (Thermo Fisher Scientific). In brief, p18-P2A-mKO2 was cloned into the pENTR2B vector (Thermo Fisher Scientific) and inserted into pROSA26-CAG-STOP-DEST, which was modified from pROSA26-STOP-DEST (Abe et al, 2011) and pCAGGS (Niwa et al, 1991). The targeted ES clones were microinjected into 8-cell stage ICR embryos, and the embryos were transferred into pseudopregnant ICR females. The resulting chimeras were bred with C57BL/6 mice, and heterozygous offspring were identified by PCR using the following primers: R26 gt Wt FW2 (P3, 5′-TCC CTC GTG ATC TGC AAC TCC AGT C-3′) and R26 gt Wt REV2 (P4, 5′-AAC CCC AGA TGA CTA CCT ATC CTC C-3′) for the wild-type allele (217 bp), and bpA FW (P7, 5′-GGG GGA GGA TTG GGA AGA CAA TAG C-3′) and R26 gt Wt REV2 for the R26R allele (297 bp). R26 FloxP CAG FW (5′-TCC TGG GCA ACG TGC TGG-3′) and R26 gt Mt FloxP REV (P5, 5′-TGT GGA ATG TGT GCG AGG CCA GAG G-3′) for the R26 allele (217 bp), R26 FloxP CAG FW and R26 gt Mt dloxP REV (P6, 5′-GCT GCA GGT CGA GGG ACC-3′) for the R26 allele (113 bp).

## Immunostaining and imaging

Brain tissues were fixed overnight in 4% paraformaldehyde at 4 °C before cryoprotection with 20% sucrose. Brains were then embedded in OCT compound for cryosectioning. After culturing in a proliferative or astrocyte induction medium, NS cells were fixed in 4% paraformaldehyde for 10 min. Frozen sections (15 μm thickness) or fixed NS cells were incubated in blocking solution (2% bovine serum albumin and 0.1% Triton X-100 in PBS), then overnight with primary antibodies (rabbit anti-AldH1l1 (1:500; #ab87117, Abcam), goat anti-TAG-1 (1:50; #4D7/TAG1, DSHB), rabbit anti-Neurogenin2 (1:200; #13144, CST), rabbit anti-GFAP (1:500; #Z0334, Dako), mouse anti-S100b (1:500; #ab11178, Abcam), mouse anti-Satb2 (1:500; #ab51502, Abcam), rabbit anti-Cux1 (1:500; #11733-1-AP, Protein-tech), rat anti-Ctip2 (1:200; #ab18465, Abcam), rat anti-PECAM (1:500; #550274, BD Pharmingen), rat anti-PDGFRa (1:500; #16-1401-82, Invitrogen), rabbit anti-SOX10 (1:500; #89356, CST), mouse anti-Dlx2 (1:500; #sc-393879, Santa Cruz), rabbit anti-Sox9 (1:500; #AB5535, Millipore), or rabbit anti-Sp8 (1:500; #NBP2-49109, Novus) and counterstained with 4,6-di-amino-2-phenyl-indole dihydrochloride (DAPI). Alexa 488, 555, and 647 nm conjugated secondary antibodies (1:1000; Molecular Probes) were then applied. To confirm p18 or p27 expression in brain tissue, rabbit anti-p18$^{Ink4c}$ (1:1000; #sc-865, Santa Cruz) or rabbit anti-p27$^{Kip1}$ (1:1000; #PA-5-27188, Invitrogen) was used, and the signal was amplified by biotin-XX-labeled tyramides (Tyramide Signal Amplification Kits; #40993, Molecular Probes). To confirm cell cycle arrest, EdU was detected using Click-iT EdU imaging kits (#C10340, Invitrogen). Briefly, sections were incubated overnight with primary antibody against EGFP to detect the electroporated cells by immunostaining. EdU detection was carried out before the secondary antibody reaction.

Samples were observed using a confocal laser microscope (Ti-RCP, Nikon and Dragonfly 502w, Andor Technology), and composite images were created using Adobe Photoshop software (Adobe Systems). Image quantifications were performed using Metamorph or ImageJ software; we first marked the number of electroporated cells per image under a 20× or 40× objective lens and then counted double-positive cell populations. Cell numbers were averaged, statistical analyses were performed, and the results are shown in a graph generated by Prism software. The total counted cell number for each condition is displayed in italics in the figure legends. The threshold and image calculator functions of ImageJ were used to quantify the overlap of immunostaining signals.

## Laser microdissection

Brain tissues were collected from E13.5 to E17.5 of *AldH1l1*-EGFP mice or E13.5 and E16.5 of NestinCreER$^{T2}$/p18mKO2 mice and then embedded in OCT compound without fixation. Fresh frozen tissue was cryosectioned at 30 μm thickness and then fixed with ice-cold 70% ethanol for 1 min. Sections washed with RNase-free water to remove OCT compound were dissected using laser microdissection microscopy (PALM MicroBeam with Axio Observer Z1, Zeiss) to collect only the ventricular surface of dorsal cortexes. Ten to 12 pieces of tissue were collected, and total mRNA was extracted to investigate p18, p27, and EGFP expression levels in *AldH1l1*-EGFP mice and bulk RNA-seq in NestinCreER$^{T2}$/p18mKO2 mice. Total mRNA was isolated using NucleoSpin RNA XS (#740902.50, Macherey-Nagel) and then applied to TapeStation 4200 (Agilent) to examine quality and measure concentrations.

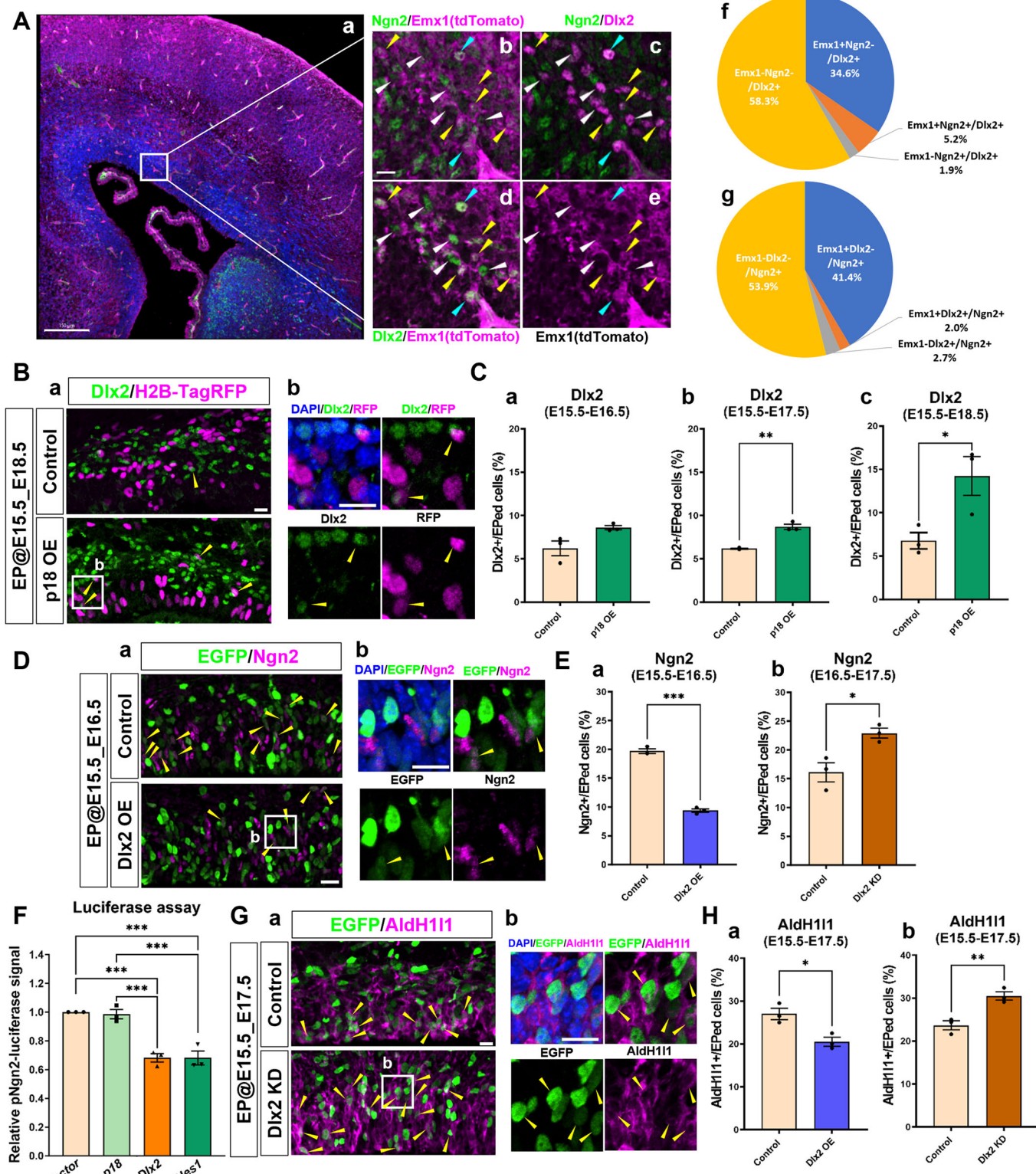

## Plasmids

Plasmids were constructed according to standard protocols for DNA cloning. Prior to electroporation into mouse brain tissue or transfection into cultured cells, plasmids were purified with the

PureLink HiPure Plasmid Maxiprep Kit (#K2100-07, Invitrogen). The following plasmids were constructed in previous studies: pCAGGS (Niwa et al, 1991); pCAG-NLS-EGFP (Konno et al, 2008); pCAG-p18 (Kosodo et al, 2011); pCAG-FloxP-EGFP-F and pCAG-Cre (Shitamukai et al, 2011); and LacZmiRNA (Kosodo et al, 2011).

**Figure 7. Expression of Dlx2 and its function in the dorsal cortex.**

(A) (a) Images of E17.5 Emx1-Cre/Ai9 mouse brains stained with Ngn2 and Dlx2 antibodies. Bar, 150 μm. (b–e) Magnified view of the square region in (a). Arrowheads; blue, Dlx2 + /Ngn2 + /tdTomato + ; yellow, Dlx2 + /Ngn2-/tdTomato + ; white, Dlx2 + /tdTomato-. Bar, 15 μm. (f, g) Quantification of the stained images (n = 4 brains each). Total counted cells of Dlx2 + , Ngn2 + ; *311, 282*. (B) Plasmids to induce OE of p18 mixed with pCAG-H2B-TagRFP were delivered at E15.5 and then fixed after 1–3 d. Sections were stained using Dlx2 antibody. (a) Images of fixation at E18.5 are presented. Arrowheads; double-positive cells to Dlx2 and H2B-TagRFP. Bar, 30 μm. (b) Magnified view of the square region in (a). Arrowheads; double-positive cells to Dlx2 and H2B-TagRFP. Bar, 10 μm. (C) Quantification of the Dlx2-positive cells among EGFP-positive cells in (B) (n = 3 brains each). Total counted cells of Control, p18 OE; *410, 549* (E16.5); *453, 413* (E17.5); *309, 167* (E18.5). Two-tailed t test; error bars show mean ± SEM. **P = 0.00118109 (b), *P = 0.03660486 (c). (D) Plasmids to induce OE and KD of Dlx2 mixed with pCAG-NLS-EGFP were delivered at E15.5 and 16.5, respectively, and then fixed after 1 d. Sections were stained using Ngn2 antibody. (a) Images of IUE at E15.5 for Dlx2 OE are presented. Arrowheads; double-positive cells for Ngn2 and EGFP. Bar, 30 μm. (b) Magnified view of the square region in (a). Arrowheads; double-positive cells to Ngn2 and EGFP. Bar, 10 μm. (E) Quantification of the Ngn2-positive cells among EGFP-positive cells in (D) (n = 3 brains each). Total counted cells of Control, Dlx2 OE; *337, 438*; Control, Dlx2 KD; *425, 322*. Two-tailed t test; error bars show mean ± SEM. ***P = 0.00003530 (a), *p = 0.03660486 (b). (F) Luciferase assay using the promoter region of Ngn2 in HEK293T cells transfected with plasmids to induce p18, Dlx2, and Hes1 expression (n = 3 independent cultures). One-way ANOVA with Tukey's test; error bars show mean ± SEM. From top to bottom, ***P = 0.00049901, ***P = 0.00069031, ***P = 0.00051001, ***P = 0.00070606. (G) Plasmids to induce OE and KD of Dlx2 mixed with pCAG-NLS-EGFP were delivered at E15.5 and then fixed at E17.5. Sections were stained using AldH1l1 antibody. (a) Arrowheads; double-positive cells for AldH1l1 and EGFP. Bar, 30 μm. (b) Magnified view of the square region in (a). Arrowheads; double-positive cells to AldH1l1 and EGFP. Bar, 10 μm. (H) Quantification of the AldH1l1-positive cells among EGFP-positive cells in (G) (n = 3 brains each). Total counted cells of Control, Dlx2 OE; *515, 468*; Control, Dlx2 KD; *649, 581*. Two-tailed t test; error bars show mean ± SEM. *P = 0.01812292 (a), **P = 0.00895741 (b). Source data are available online for this figure.

HA-HES1 and E1-pNgn2-luc-3′UTR were constructed in previous works (Kobayashi et al, 2015; Shimojo et al, 2008). PB-*Gfa2*-EGFP (Hamabe-Horiike et al, 2021) was used with the PiggyBac Transposon Vector System (#PB210 and #PB512, System Bioscience) for the transposon-based fluorescent reporter assay. To generate pCAG-p18 (human), a full-length human *P18* coding sequence was amplified from human iPSC-derived cDNA using PCR and ligated into pCAGGS. To generate pCAG-H2B-TagRFP, the TagRFP coding sequence from pTagRFP-N (#FP142, Evrogen) was inserted into pCAG-H2B-EGFP (Konno et al, 2008) by replacing EGFP using BamH1 and Not1. To generate pCAG-p18-P2A-mKO2, first, the p18 coding sequence was ligated into pCAG-EGFP-N1 (Konno et al, 2008). Then, the mKO2 coding sequence amplified from pmKO2-S1 (#AM-V0141M, MBL) using primers containing the P2A sequence (Kim et al, 2011) was inserted by replacing EGFP using BamH1 and Not1. To generate pCAG-p27, a full-length mouse p27 coding sequence was amplified from embryonic mouse brain-derived cDNA and ligated into pCAG-CreER^{T2} (Matsuda and Cepko, 2007) using EcoR1 and Not1. To generate pCAG-Dlx2, the full-length mouse Dlx2 coding sequence was amplified from embryonic mouse brain-derived cDNA and ligated into pCAG-EGFP-N1 using EcoR1 and Not1. To generate KD constructs for p18, p27, and Dlx2, the BLOCK-iT Pol II miR RNAi Expression Vector Kit (#K4935-00, Invitrogen) was used with the listed oligos as described in a previous study (Kosodo et al, 2011). To generate shRNA constructs for p18 and p27, pSilencer 3.0-H1 (#V012700, NovoPro) was used with the listed oligos. Briefly, oligonucleotides corresponding to the target coding sequence and its complementary sequence were inserted into the pSilencer 3.0-H1 vector using *Bam*H1 and *Hind*III sites. Primer sequences are provided in Table 1.

## Quantitative real-time PCR (qPCR)

The isolated total mRNA was used to synthesize equal amounts of cDNA to quantify specific gene expression using the ReverTra Ace qPCR RT kit (#FSQ-201, TOYOBO). qPCR was performed using FastStart Universal SYBR Green Master (#0913850001, Roche) at 95 °C for 10 min, then 40 cycles of 95 °C for 15 s and 60 °C for 60 s. GAPDH was amplified as an internal control, and the specificity of the reaction was verified by melt curve analysis. Experiments were

conducted using a Light Cycler 480II (Roche). Primer sequences used for quantitative real-time PCR are provided in Table 2.

## Preparation of NS cells

NS cells were prepared according to a previous publication (Imayoshi et al, 2013) with some modifications. Dorsal cortexes were dissected from EGFP-positive brains of E14.5 *AldH1l1*-EGFP mice and enzymatically dissociated using papain (#LK003153, Worthington) in DMEM/F12 medium. Cells were plated on a 12-well plate coated with 15 μg/ml poly-ʟ ornithine (#P3655, Sigma) and 1 μg/ml fibronectin (#1030-FN-05M, R&D Systems). The cell density was adjusted to $2.5 \times 10^5$ cells/well and incubated at a confluence of 50–60% in a 12-well plate for transfection. One microgram of DNA from the mKO2 or p18-2A-mKO2 construct was mixed with 3 μl of ViaFect transfection reagent (#E4982, Promega) for p18-overexpression experiments. The mixture was incubated in NS cell culture medium for 10 h. The cells were washed with PBS, and then proliferative medium (DMEM/F12 (#11039-021, Gibco), 1× penicillin/streptomycin (#15140-122, Gibco), 1× N2 plus media supplement (#AR009, R&D Systems), 20 ng/ml bFGF (#060-04543, Wako), 20 ng/ml EGF (#53003-018, Invitrogen)) or astrocyte induction medium (DMEM/F12, 0.5× penicillin/streptomycin, 1× N2 plus media supplement, 1× B27 (#12587-010, Invitrogen), 80 ng/ml LIF (#ESG1106, Millipore), 80 ng/ml BMP4 (#314-BP-010, R&D Systems), FBS (#16000-044, Gibco)) were added. Cells were fixed for immunostaining after 48 h.

## Flow cytometry and cell sorting

Electroporated NS cells were incubated in astrocyte induction medium for 48 h, harvested by centrifugation, and resuspended in PBS. Subsequently, only electroporated (RFP-positive) cells were isolated by an automated cell sorter (MoFlo Astrios, Beckman Coulter) with 555 nm excitation/584 nm emission wavelengths, and total mRNA was extracted for qPCR from the sorted cell populations.

## In utero electroporation

Pregnant mice were anesthetized with 10% sodium pentobarbital, and their intrauterine embryos were exposed after cesarean section.

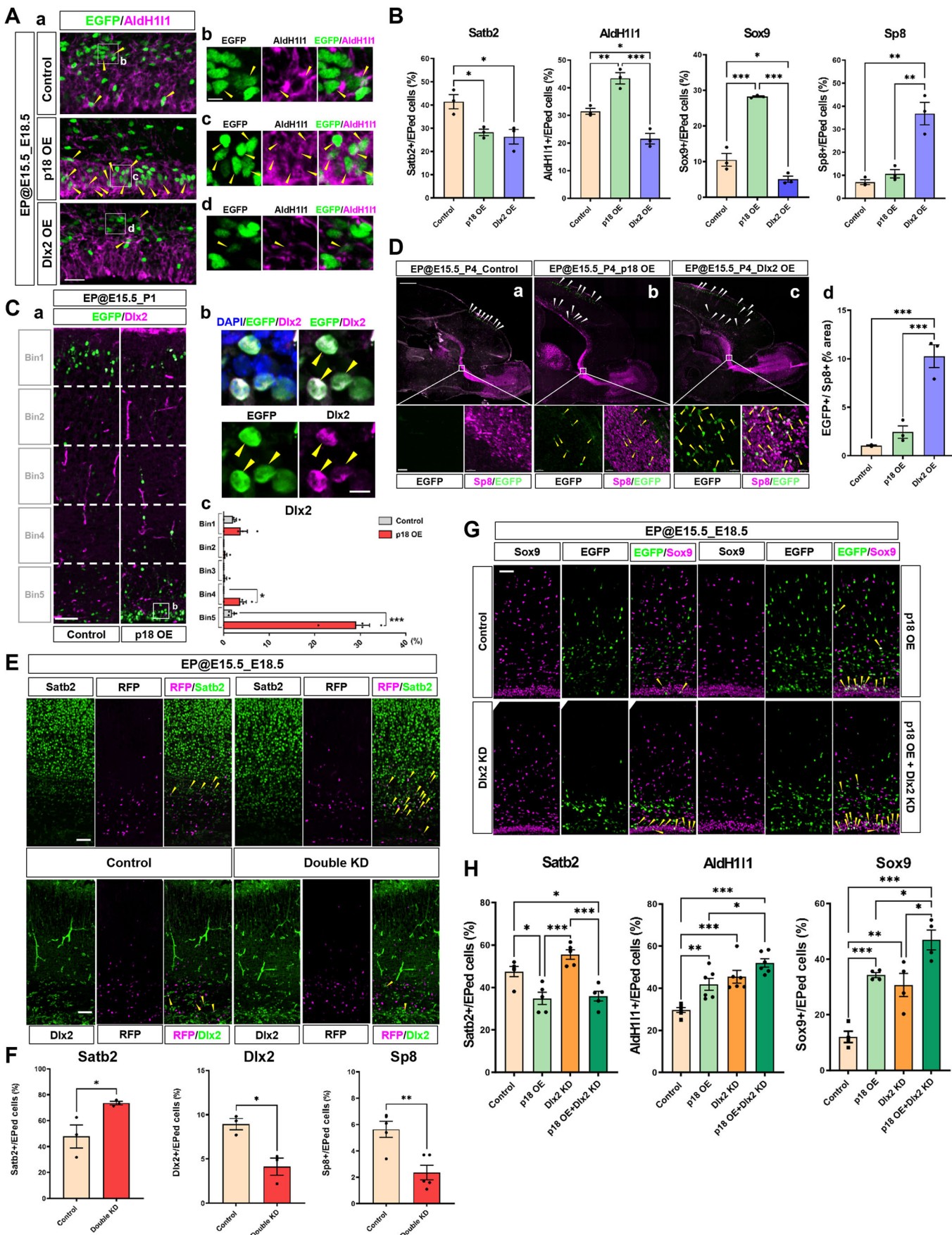

◄ **Figure 8. Lineage transitions of UL neurons, astrocytes, and OB interneurons.**

(A) Plasmids to induce OE of p18 and Dlx2 mixed with pCAG-NLS-EGFP were delivered at E15.5 and then fixed at E18.5. Sections were stained using Satb2, AldH1l1, Sox9, and Sp8 antibodies. (a) Images of staining with AldH1l1 are presented. Bar, 50 µm. (b–d) Magnified view of the square region in (a). Bar, 10 µm. Arrowheads; double-positive cells for AldH1l1 and EGFP. (B) Quantification of the Satb2-, AldH1l1-, Sox9-, and Sp8-positive cells among EGFP-positive cells in (A) ($n = 3$ brains each). Total counted cells of Control, p18 OE, Dlx2 OE; *283, 261, 380* (Satb2, SP8); *324, 256, 328* (AldH1l1); *340, 249, 265* (Sox9). One-way ANOVA with Tukey's test; error bars show mean ± SEM. From top to bottom, left to right, *$P = 0.02873290$, *$P = 0.01608185$ (Satb2), *$P = 0.01522693$, **$P = 0.00652441$, ***$P = 0.00026211$ (AldH1l1), *$P = 0.03799166$, ***$P = 0.00009784$, ***$P = 0.00002040$ (Sox9), **$P = 0.00113463$, **$P = 0.00226063$ (Sp8). (C) (a) Plasmids to induce OE of p18 mixed with pCAG-NLS-EGFP were delivered at E15.5, followed by fixation at P1. Sections were stained using Dlx2 antibody. Bar, 100 µm. Arrowheads; double-positive cells for Dlx2 and EGFP. (b) Magnified view of the square region in (a). Bar, 10 µm. (c) The bars indicate the proportions of Dlx2-positive cells among electroporated cells in each BIN. $t$ tests were performed between the control and p18 OE in the same BINs ($n = 3$ brains each). Total counted cells of Control, p18 OE; *295, 246*. Two-tailed $t$ test; error bars show mean ± SEM. From top to bottom, *$P = 0.01566438$, ***$P = 0.00004670$. (D) (a–c) Plasmids to induce OE of p18 and Dlx2 were delivered under the same conditions as C and then fixed at P4. Sections were stained using Sp8 antibody to detect RMS (magnified in the bottom). White arrowheads; EGFP + /Sp8- cells. Yellow arrowheads; EGFP + /Sp8+ cells. Bars in low and high magnified view, 500 and 20 µm, respectively. (d) Quantification of the EGFP-positive area in (a–c) ($n = 3$ brains each, normalized by the Sp8-positive area). One-way ANOVA with Tukey's test; error bars show mean ± SEM. From top to bottom, ***$P = 0.00092368$, ***$P = 0.00037638$. (E) Plasmids to induce KD of p18 and p27 mixed with pCAG-H2B-TagRFP were delivered at E15.5 and then fixed at E18.5. Sections were stained using Satb2, Dlx2, and Sp8 antibodies. Images of staining with Satb2 and Dlx2 are presented. Arrowheads; double-positive cells for Satb2 and EGFP. Bar, 100 µm. (F) Quantification of Satb2-, Dlx2-, and Sp8-positive cells among H2B-TagRFP-positive cells in (E) ($n = 3$–5 sections from 3 brains each). Total counted cells of Control, p18/p27 KD; *179, 159* (Satb2); *188, 237* (Dlx2); *428, 376* (SP8). Two-tailed $t$ test; error bars show mean ± SEM. *$P = 0.04605735$ (Satb2), *$P = 0.01445994$ (Dlx2), **$P = 0.00401234$ (Sp8). (G) Plasmids to induce OE of p18 and KD of Dlx2 mixed with pCAG-NLS-EGFP were delivered at E15.5 and then fixed at E18.5. Sections were stained using Satb2, AldH1l1, and Sox9 antibodies. Images of staining with Sox9 are presented. Arrowheads; double-positive cells for Sox9 and EGFP. Bar, 100 µm. (H) Quantification of Satb2-, AldH1l1-, and Sox9-positive cells among EGFP-positive cells ($n = 4$–6 sections from 3 brains each). Total counted cells of Control, p18 OE, Dlx2 KD, p18 OE with Dlx2 KD; *449, 326, 280, 272* (Satb2); *345, 262, 222, 236* (AldH1l1); *411, 432, 331, 394* (Sox9). One-way ANOVA with Tukey's test; error bars show mean ± SEM. From top to bottom, left to right, *$P = 0.02158542$, *$P = 0.01150958$, ***$P = 0.00012278$, ***$P = 0.00022252$ (Satb2), ***$P = 0.00001046$, *$P = 0.03252156$, ***$P = 0.00077312$, **$P = 0.00848468$ (AldH1l1), ***$P = 00001319$, *$P = 0.04602633$, *$p = 0.01037045$, **$P = 0.00386438$, ***$P = 0.00094264$ (Sox9). Source data are available online for this figure.

Approximately 1 µl of the plasmid solution containing 0.01% fast green was injected into the brain ventricle using a mouth-controlled micropipette. Electronic pulses (40 V, 50 ms, 950-ms intervals, 4 times) were then applied using an electroporator (NEPA21, NEPAGENE). The concentrations of the plasmid were as follows: 1.0 µg/µl pCAG-p18, pCAG-p27, pCAG-Dlx2, and pCAG-empty vector (for negative control); 0.5 µg/µl pCAG-H2B-TagRFP for overexpression; 1.0 µg/µl miR p18, miR p27, miR Dlx2, and miR LacZ (for negative control); 0.5 µg/µl pCAG-H2B-TagRFP and pCAG-NLS-EGFP for knockdown; and 1.0 µg/µl PB-Gfa2-EGFP, PB-RFP, and 0.5 µg/µl PB Transposase. All plasmids were delivered to the dorsal cortex.

## Luciferase assay

A luciferase assay was performed using the Dual-Luciferase Reporter Assay System (#E1910, Promega) according to the company's instructions. Briefly, HEK293T cells were cultured in DMEM with high glucose (#LM 001-09, Welgene), 10% FBS, and 1× penicillin/streptomycin. Plasmids were transfected using PEI-MAX (#24765-1, Polysciences) in Opti-MEM (#31985-062, Invitrogen). For normalization, pRL-TK plasmids (Promega) were cotransfected. After 24 h of culture, cell lysates were prepared using a Passive Lysis Buffer kit. Luciferase Assay Buffer II and Stop and Glo Substrate were applied sequentially during the quantification of luminescent signals using a FlexStation 3 microplate reader system (Molecular Probes).

## DNA methylation analysis

After making fresh frozen sections using NestinCreER^T2/p18mKO2 E13.5 and E16.5 mouse embryonic brains treated with tamoxifen at E12.5 and E15.5, respectively, the target area, including the ventricular zone, was dissected and collected using laser micro-dissection. Their genomic DNA was extracted using the NucleoSpin Tissue XS kit. Sodium bisulfite treatment of genomic DNA was performed using the MethylEasy™ Xceed Rapid DNA Bisulfite Modification Kit (Genetic Signatures). The region of the bisulfite-treated Gfap promoter (Takizawa et al, 2001) was amplified by PCR using the following primers: GFAP_Tak_F2 (5′-GTATAGA-TATAATGGTTAGGGGTG-3′) and GFAP_Tak_R (5′-TCTAC CCATACTTAAACTTCTAATATCTAC-3′). The PCR products were cloned into a TA cloning vector (RBC Bioscience Corp.), and 10-12 randomly picked from each of 3–4 independent PCR amplifications from different mouse brains were sequenced.

## Morphological analysis of 3D-reconstructed cells

To visualize the single-cell morphology, we used a sparse expression vector system described previously (Shitamukai et al, 2011). Embryonic brains were coelectroporated with pCAG-FloxP-EGFP-F (0.5 µg/µl) and a low concentration of Cre recombinase expression vector (10 ng/µl) with 0.5 µg/µl pCAG-H2B-TagRFP, which resulted in high and sparse expression of fluorescent reporters labeling the cellular membrane and nuclear position by EGFP and TagRFP, respectively. Electroporated brains were dissected at the P10 stage and fixed overnight in 4% paraformaldehyde. The fixed brains were embedded in 2% agarose in PBS and solidified on ice. The agarose block containing the brain was then glued to a metal mounting block and sectioned at 200 µm thickness using a vibratome (VT1200S, Leica). Images of sparsely labeled electroporated cells in the brain were acquired using an inverted confocal microscope (Ti-RCP, Nikon). A 20x objective lens (NA = 0.75) was used with a zoom factor of 6.410. Fifty-micron z-stack sections were acquired with a 0.6 µm interval between each optical section.

Subsequently, stacked images were subjected to quantitative morphological analysis using MATLAB, performed by another examiner without knowing the contents of electroporated plasmids (blind test). To quantify nuclear volume, a signal of the RFP channel was extracted and analyzed. A median and average filter was applied ($3 \times 3 \times 3$). Subsequently, binarization was performed

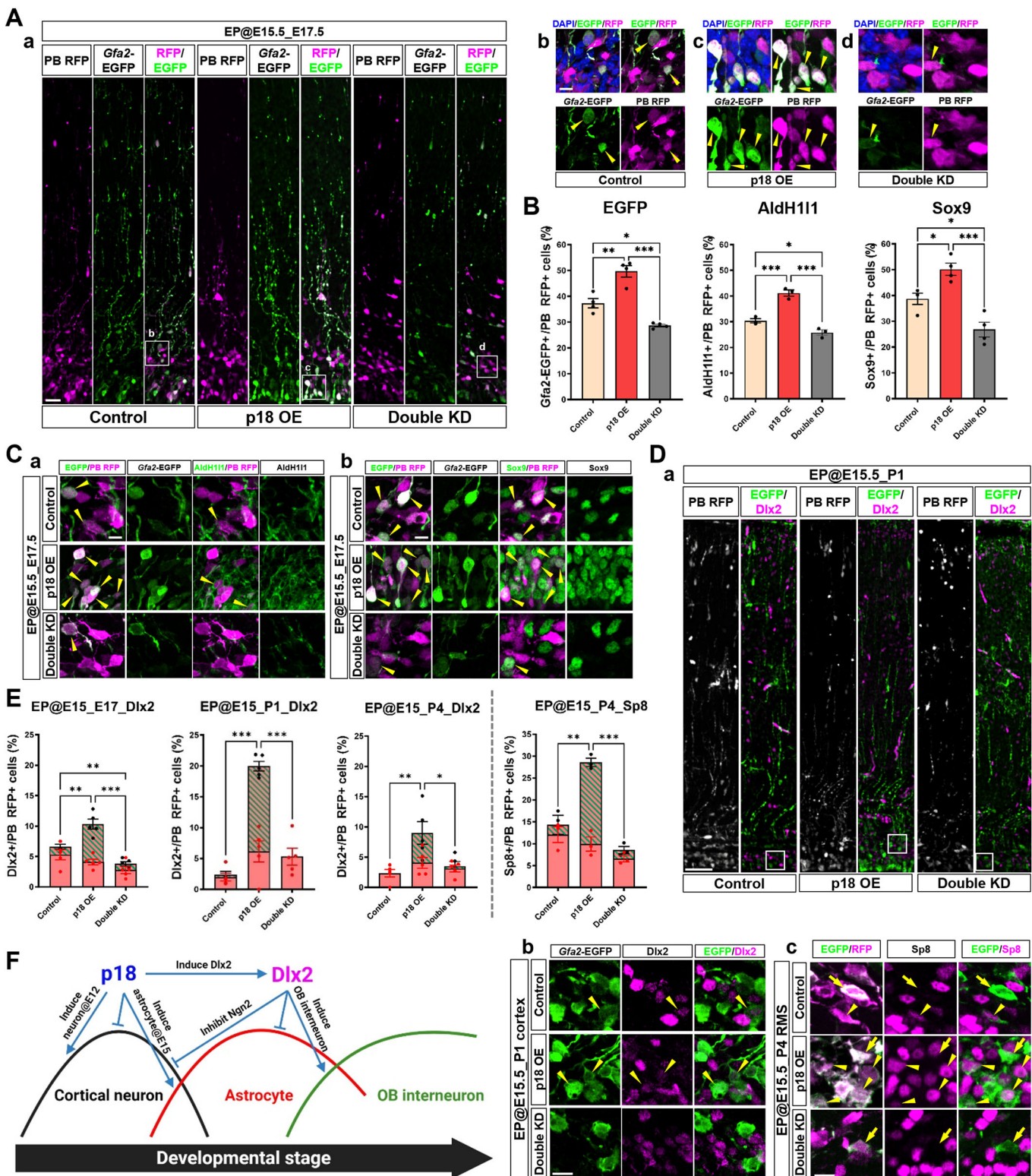

using the threshold of 50% of the signal-to-noise ratio. Finally, the centroid position (Xc, Yc, and Zc) was calculated. For the analysis of the cellular process occupied area, a signal of the EGFP channel projected to the XY-plane was extracted. Averaging and binarization were performed as indicated for the RFP channel. Blank

among processes was filled after maximum intensity projection. The farthest position from the centroid (Xc, Yc, and Zc) in the projected area was identified. The image field was rotated to fit the point on the y-axis, and then the projected area was subdivided into 8 orientations. Based on this image processing, the length of the

◀

**Figure 9. Investigation of the lineage transitions at the single-cell level.**

(A) (a) Plasmids to induce OE of p18 and double KD of p18/p27 mixed with PB-*Gfa2*-EGFP, PB-RFP, and PBase were delivered at E15.5 and then fixed at E17.5. Sections were stained using AldH1l1 or Sox9 antibody. Images of EGFP and RFP are presented. Bar, 30 μm. (b–d) Magnified view of the square region in (a). Arrowheads; double-positive cells for EGFP and RFP. Bar, 10 μm. (B) Quantification of EGFP-, AldH1l1-, and Sox9-positive cells among RFP-positive cells ($n = 3$–4 sections from 3 brains each). Total counted cells of Control, p18 OE, Double KD; *234, 298, 349* (EGFP and Sox9); *146, 177, 207* (AldH1l1). One-way ANOVA with Tukey's test; error bars show mean ± SEM. From top to bottom, left to right, *$P = 0.01406736$, **$P = 0.00152748$, ***$P = 0.00002786$ (EGFP), *$P = 0.04733737$, ***$P = 0.00086532$, ***$P = 0.00011757$ (AldH1l1), *$P = 0.02053167$, *$P = 0.02463277$, ***$P = 0.00025819$ (Sox9). (C) Images of staining with AldH1l1 (a) and Sox9 (b) antibodies. Arrowheads; double-positive cells for AldH1l1 or Sox9 and RFP. Bar, 10 μm. (D) (a) Plasmids to induce OE of p18 and double KD of p18/p27 mixed with PB-*Gfa2*-EGFP, PB-RFP, and PBase were delivered at E15.5 and then fixed at P1. Sections were stained with Dlx2 antibody. Bar, 50 μm. (b) Magnified view of the square region in (a). Arrowheads; double-positive cells for EGFP and Dlx2. Bar, 10 μm. (c) Images of the RMS at P4 stained with Sp8 antibody. Arrowheads; double-positive cells for EGFP and Sp8. Arrows; EGFP-positive and Sp8-negative cells. Bar, 10 μm. (E) Quantification of Dlx2-positive cells in the cortex and Sp8-positive cells in the RMS among RFP-positive cells ($n = 3$–4 sections from 3 brains each). Each bar is split into *Gfa2*-EGFP-positive (striped area) and -negative (solid area) cells. Total counted cells of Control, p18 OE, Double KD; *280, 342, 409* (Dlx2, E17.5); *410, 343, 309* (Dlx2, P1); *200, 270, 327* (Dlx2, P4); *181, 191, 178* (Sp8, P4). One-way ANOVA with Tukey's test; error bars show mean ± SEM. From top to bottom, left to right, **$P = 0.00964913$, **$P = 0.00150298$, ***$P = 0.00000773$ (E17.5, Dlx2), ***$P = 0.00000006$, ***$P = 0.00000044$ (P1, Dlx2), **$P = 0.00616579$, *$P = 0.02057649$ (P4, Dlx2), **$P = 0.00108087$, ***$P = 0.00016285$ (P4, Sp8). (F) Schematics of the lineage transition regulated by p18. Source data are available online for this figure.

**Table 1. Primers for plasmids generated in this study.**

| Construct | Primer name | Sequences |
|---|---|---|
| pCAG-p18 (human) | p18_Human EcoA2nd | GGGAATTCCACCATGGCCGAGCCTTGGGGG |
| | p18_Human BamB2nd | CCGGATCCTTATTGAAGATTTGTGGCTCCCCC |
| pCAG-p18-P2A-mKO2 | N1p18EcoA | GGGAATTCCACCATGGCCGAGCCTTG |
| | N1p18BamB | CCGGATCCTGCAGGCTTGTGGCTCCC |
| | P2AmKO2N1BamH1A | CCGGATCCAGGAAGCGGAGCTACTAACTTCAGCCTGCTGAAGCAGGCTGGAGACGTGGAGGAGAACCCTGGACCTATGGTGAGTGTGATTAAACC |
| | P2AmKO2N1Not1B | GGGCGGCCGCTTAGGAATGAGCTACTGCATC |
| pCAG-p27 | p27_mouse EcoA | GGGAATTCACCATGTCAAACGTGAGAGTGTCTAACGG |
| | p27_mouse Not1B | CCGCGGCCGCTTACGTCTGGCGTCGAAGGCC |
| pCAG-Dlx2 | mDlx2_EcoA | CCGAATTCAGGATGACTGGAGTCTTTGACAG |
| | mDlx2_Stop_Not1B | CCGCGGCCGCTTAGAAAATCGTCCCCGCGCTC |
| p18 miRNA | p18_816Top | TGCTGAAACAAAGCGAAAGGAAACCCGTTTTGGCCACTGACTGACGGGTTTCCTCGCTTTGTTT |
| | p18_816Bot | CCTGAAACAAAGCGAGGAAACCCGTCAGTCAGTGGCCAAAACGGGTTTCCTTTCGCTTTGTTTC |
| p27 miRNA | p27_1449_TOP | TGCTGACAAATTGAAGCAAGTTCTTCGTTTTGGCCACTGACTGACGAAGAACTCTTCAATTTGT |
| | p27_1449_BOT | CCTGACAAATTGAAGAGTTCTTCGTCAGTCAGTGGCCAAAACGAAGAACTTGCTTCAATTTGTC |
| Dlx2 miRNA | Dlx2_986A | TGCTGTTCCACATCTTCTTGAACTTGGTTTTGGCCACTGACTGACCAAGTTCAAAGATGTGGAA |
| | Dlx2_986B | CCTGTTCCACATCTTTGAACTTGGTCAGTCAGTGGCCAAAACCAAGTTCAAGAAGATGTGGAAC |
| p18 shRNA | 816 A #1 | GATCCGGTTTCCTTTCGCTTTGTTTCAAGAGAAACAAAGCGAAAGGAAACCTTTTTTGGAAA |
| | 890 A #2 | GATCCAGTTTATGAAATATTTAAATTCAAGAGATTTAAATATTTCATAAACTTTTTTTGGAAA |
| | 925 A #3 | GATCCAAAATTCTGATTTCTAACATTCAAGAGATGTTAGAAATCAGAATTTTTTTTTTGGAAA |
| | 942 A #4 | GATCCCATGTAATAGCTATTCCTTTTCAAGAGAAAGGAATAGCTATTACATGTTTTTTGGAAA |
| p27 shRNA | 1449 A #1 | GATCCAGAACTTGCTTCAATTTGTTTCAAGAGAACAAATTGAAGCAAGTTCTTTTTTTGGAAA |
| | 1844A #2 | GATCCGAAAAATTCTTATTTCTTTTTCAAGAGAAAGAAATAAGAATTTTTCTTTTTTGGAAA |
| | 1907A #3 | GATCCGTCTGTAACTTTACACAAATTCAAGAGATTTGTGTAAAGTTACAGACTTTTTTGGAAA |
| | 2375A #4 | GATCCGAAAACTTCCGTAGTTATTTTCAAGAGAAATAACTACGGAAGTTTTCTTTTTTGGAAA |

longest process and cellular process area projected to the XY-plane were calculated. For the analysis of the cellular process number, a signal of the EGFP channel was treated by a Gaussian filter to enhance the process structure. After binarization using the threshold set to 50% of the signal-to-noise ratio, line extraction was performed in three-dimensional volume. Subsequently, projection to the XY plane, image field rotation, and area division into 8 orientations were conducted. Based on this image processing, the number and density of processes were calculated. Sholl analysis was further performed for the projected image to count the number of

**Table 2. Primers for quantitative real-time PCR.**

| Gene | Primer | Sequences |
|---|---|---|
| GAPDH | Forward | CACTCACGGCAAATTCAACGG |
| | Reverse | ACTCCACGACATACTCAGCACC |
| EGFP | Forward | AGGACGACGGCAACTACAAGAC |
| | Reverse | AGTTGTACTCCAGCTTGTGCCC |
| p18$^{Ink4c}$ | Forward | ATGCTGCCAGAGCAGGTTTC |
| | Reverse | ACATTGCAGGCTGTGTGCTTC |
| p27$^{Kip1}$ | Forward | AGCTGAGAGTGTCTAACGGGAG |
| | Reverse | TTCATGATTGACCGGGCCGAAG |
| Dlx2 | Forward | CGCACCATCTACTCCAGTTTCC |
| | Reverse | TCTCGCCGCTTTTCCACATC |
| Neurogenin2 | Forward | TGCAGCGCATCAAGAAGACC |
| | Reverse | GATCTTCGTGAGCTTGGCATCC |
| AldH1l1 | Forward | TCTCCAAATGCCCTACCAGCTC |
| | Reverse | ACCGCCTTGTCAACATCACTC |
| GFAP | Forward | GAGAGAGATTCGCACTCAGTA |
| | Reverse | TGAGGTCTGCAAACTTGGAC |
| Satb2 | Forward | CCAATGTGTCAGCAACCAAG |
| | Reverse | CTCTCGCTCCACTCTTTCCA |

processes across the radius of 6 to 80 μm from the centroid (2 μm steps).

## Statistical analysis

Prism (GraphPad Software) was used for statistical analyses. Differences between two groups were analyzed by two-tailed Student's $t$ test, and differences between more than two groups were analyzed by ANOVA followed by Tukey's test. Differences were considered significant for $P$ values <0.05 (*$P < 0.05$, **$P < 0.01$, and ***$P < 0.001$) unless specifically indicated. All error bars represent the SEM. For qPCR analysis, each point indicates the averaged value of triplicate reactions.

## mRNAseq and computational analysis

mRNA-seq was performed on tissue fragments of the VZ in the dorsal cortex cut by LMD. All mRNA-seq libraries were generated using the NEXTflex Rapid Directional RNA-Seq Kit (#NOVA-5198-10, PerkinElmer) with RNA-Seq Barcodes 1–24 beads. The RNA-seq read data were obtained using an Illumina HiSeq 2500.

The adapter sequences were trimmed by cutadapt v3.4. Paired-end reads were aligned to the mouse transcriptome GRCm38 (Ensembl release 100) by STAR v2.7.2b with ENCODE standard options (Dobin et al, 2013). The quantity of aligned read count was estimated by RSEM v1.2.31 (Li and Dewey, 2011). Genes with total read counts less than 10 were excluded from further analysis. The false discovery rate (FDR; Benjamini-Hochberg procedure) from differential expression analysis for each gene was calculated by DESeq2-1.30.1 (Love et al, 2014). Differentially expressed genes were defined with the following threshold: fold change >2 and FDR < 0.05. The functional enrichment analysis of differentially expressed genes was performed by Metascape (Zhou et al, 2019).

The single-cell RNA-seq data of the developing mouse dorsal forebrain were collected from previous research (La Manno et al, 2021). Each cell type and subtype were annotated by their marker genes. The deconvolution of each RNA-seq data point was performed by BayesPrism v2.0 using the scRNA-seq count matrix as a reference (Chu et al, 2022).

## Data availability

The RNA-seq data set is deposited to Gene Expression Omnibus, GEO database accession number: GSE265784. The source data are available online for this paper. All unique materials generated in this study are available from the lead contact or deposited organizations with a completed materials transfer agreement.

The source data of this paper are collected in the following database record: biostudies:S-SCDT-10_1038-S44318-024-00325-9.

## Peer review information

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

## Acknowledgements

The authors thank Ryoichiro Kageyama, Hiromi Shimojo, Itaru Imayoshi, Fumio Matsuzaki, Daijiro Konno, and Hiroshi Kawasaki for providing research materials; Katsuhiko Ono, Gwanghyun Park, Youngjae Ryu and Sophors Phorl for valuable supports and discussions; and the KBRI Research Equipment Core Facility Team for technical assistance. This study was funded by grants from the KBRI Basic Research Program of the Ministry of Science, ICT, Future Planning (24-BR-01-03, 24-BR-03-01 to YK) and the National Research Foundation of Korea (2022R1A2C1002728 to YK; 2022R1A2C3011663 to TYR).

## Author contributions

**Wonyoung Lee**: Conceptualization; Data curation; Software; Formal analysis; Investigation; Visualization; Methodology; Writing—original draft; Writing—review and editing. **Byunghee Kang**: Data curation; Software; Formal analysis; Writing—original draft. **Hyo-Min Kim**: Data curation; Software; Formal analysis. **Tsuyoshi Ishida**: Investigation. **Minkyung Shin**: Formal analysis; Investigation; Methodology. **Misato Iwashita**: Data curation; Formal analysis; Investigation; Visualization; Methodology. **Masahiro Nitta**: Investigation. **Aki Shiraishi**: Investigation. **Hiroshi Kiyonari**: Investigation. **Koichiro Shimoya**: Investigation. **Kazuto Masamoto**: Formal analysis; Methodology. **Tae-Young Roh**: Supervision; Validation; Investigation. **Yoichi Kosodo**: Conceptualization; Supervision; Funding acquisition; Validation; Investigation; Writing—original draft; Project administration; Writing—review and editing.

Source data underlying figure panels in this paper may have individual authorship assigned. Where available, figure panel/source data authorship is listed in the following database record: biostudies:S-SCDT-10_1038-S44318-024-00325-9.

## Disclosure and competing interests statement

The authors declare no competing interests.

# Expanded View Figures

**Figure EV1.   The expression level of CDKIs, confirmation of markers, and sample preparation by laser microdissection (LMD).**  ▶

(**A**) The expression level of CDKIs in RGCs in the developing mouse cortex (reanalysis of La Manno et al, 2021). The dot plot represents the expression level of CDKIs in the population of RGCs from La Manno et al scRNA-seq data. The dot size and color correspond to the proportion of cells that express the gene and the average level (log2-normalized count) of expressing cells. Arrows indicate higher expression at the onset of the gliogenesis stage (around E15-16). (**B**) Confirmation of the specificity of p18 and p27 antibodies. The control vector (a, b, g, h), p18 (c, d, i, j), and p27 (e, f, k, l) were transfected into mouse embryonic fibroblasts (MEFs). Subsequently, immunostaining using p18 (a-f) and p27 (g-l) antibodies was performed on the fixed cells to confirm that there were no cross-reactions between p18 and p27. Bar, 20 μm. (**C**) Sections of the embryonic mouse brains at E14.5 (a–c) and E15.5 (d–f) were immunostained with p18 and PECAM (a marker of a blood vessel) antibodies to confirm that bright dot structures responding to the p18 antibody were blood cells (arrowheads). Bar, 50 μm. (**D**) Confirmation of the specificity of the *Aldh1l1*-EGFP and AldH1l1 antibodies. (a–i) Sections of the *AldH1l1*-EGFP mouse brains at E17.5 (a–c), P0 (d–f), and P10 (g–i) were immunostained with AldH1l1 antibody to confirm that *AldH1l1*-EGFP expression reflects endogenous AldH1l1 protein expression. Bar, 50 μm. (j) Quantification of the immunostaining ($n = 3$ brains each). One-way ANOVA with Tukey's test; error bars show mean ± SEM. No statistically significant differences were detected. (**E**) LMD from mouse brain tissue. (a) Images before and after LMD of freshly frozen mouse brain sections at E13.5–17.5. (b) Quality check of RNA using TapeStation after preparation from mouse brain tissue by LMD (E13.5–17.5).

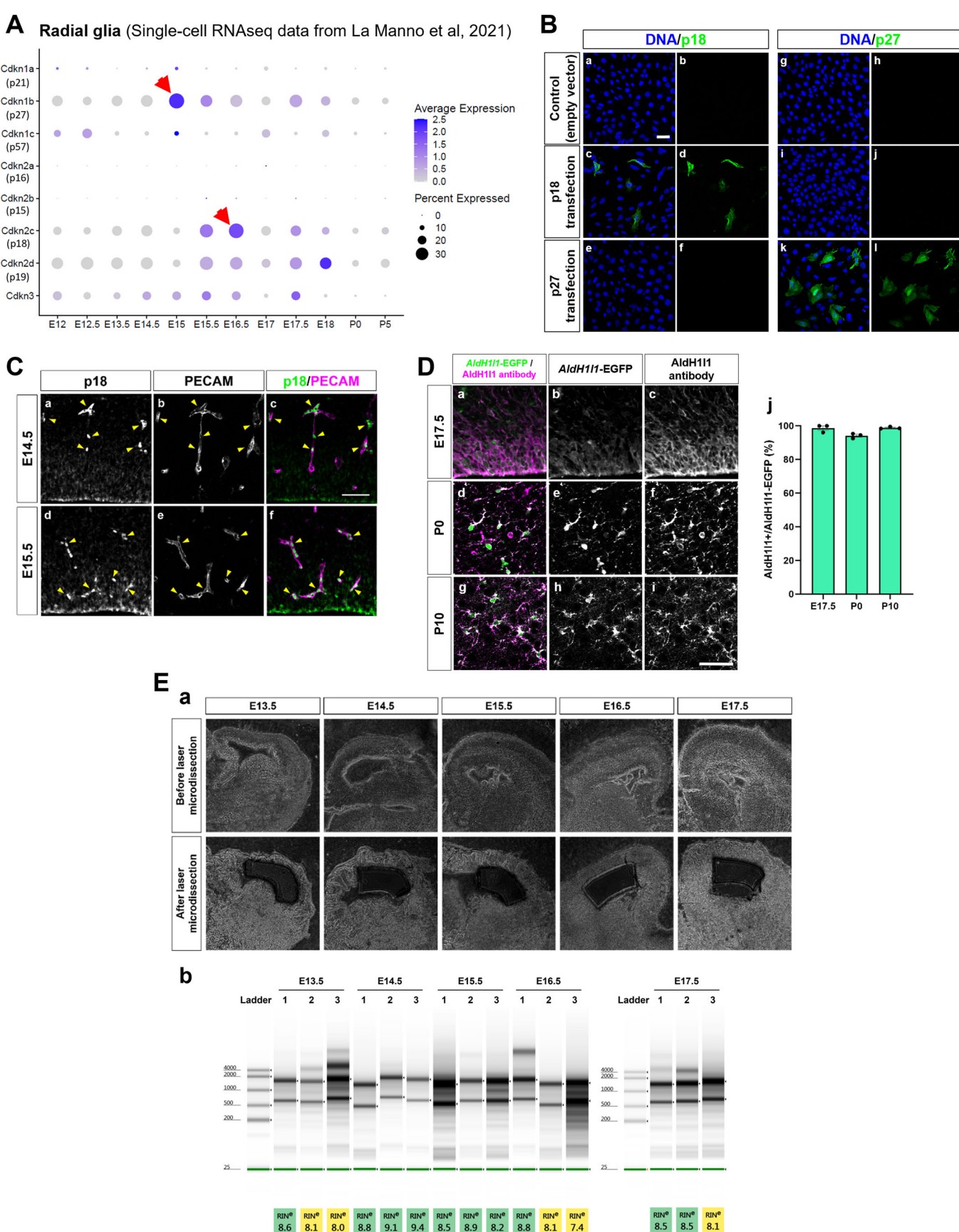

**Figure EV2.  Characterizations of p18 and p27 OE.**

(**A**) Cell cycle inhibition in p18 and p27 OE brains. (a) Plasmids to induce OE of p18 and p27 mixed with pCAG-H2B-EGFP were delivered at E14.5. EdU was administered 2 h before fixation at E15.5. White squares (i-vi) are magnified images of I-III. Arrows; the border between electroporated and nonelectroporated regions. Bars, 200 μm in I-III, and 100 μm in ii, iv, and vi. (b) Quantification of EdU-positive cells among H2B-EGFP-positive cells ($n = 3$ brains each). One-way ANOVA with Tukey's test; error bars show mean ± SEM. From top to bottom, ***$P = 0.00000039$, ***$P = 0.00000120$. (**B**) Distribution of the *AldH1l1*-EGFP-positive cells with p18 OE. (a–d) Plasmids to induce OE of p18 mixed with pCAG-H2B-TagRFP were delivered at E15.5 of the *AldH1l1*-EGFP mice and then fixed at E16.5 (a), 17.5 (b), 18.5 (c), and P0 (d). Positions of the H2B-TagRFP-positive cells from the ventricular surface were measured. The bars indicate the ratio of H2B-TagRFP-positive cells in each zone among all zones ($n = 3$ brains each). Two-tailed $t$ test; error bars show mean ± SEM. From top to bottom, **$P = 0.00160483$ (c), **$P = 0.00782095$, *$P = 0.01564190$ (d). (e) Representative images of the entire span of the dorsal cortex used for the quantification in (d (P0)). The cropped images close to the ventricular surface are presented in the main Fig. 2F (P0). Bar, 50 μm. (**C**) Expressions of astrocyte markers (Sox9 and GFAP) near the ventricular surface by p18 OE. Plasmids to induce OE of p18 mixed with pCAG-H2B-TagRFP were delivered at E15.5 and then fixed at P0. Sections were stained using Sox9 (a) or GFAP (b) antibody. Arrowheads; double-positive cells to Sox9 (a) or GFAP (b) and H2B-TagRFP. Bar, 10 μm. Sox9 (c) or GFAP (d) -positive cells among H2B-TagRFP-positive cells were quantified ($n = 3$ brains). Two-tailed $t$-test; error bars show mean ± SEM. **$p = 0.00288849$ (c), **$P = 0.00379764$ (d). (**D**) Effects of p18 OE on oligodendrocyte differentiation. (a) Plasmids to induce OE of p18 mixed with pCAG-H2B-TagRFP were delivered at E15.5 of the *AldH1l1*-EGFP mice and then fixed at P0. Sections were stained using PDGFR-alpha and Sox10 antibodies to recognize the oligodendrocyte lineage. Bar, 50 μm. (b) Quantification of Sox10-, PDGFR-alpha-, and *AldH1l1*-EGFP-positive cells among H2B-TagRFP-positive cells ($n = 3$–4 brains each). Two-tailed $t$ test; error bars show mean ± SEM. **$P = 0.00153132$.

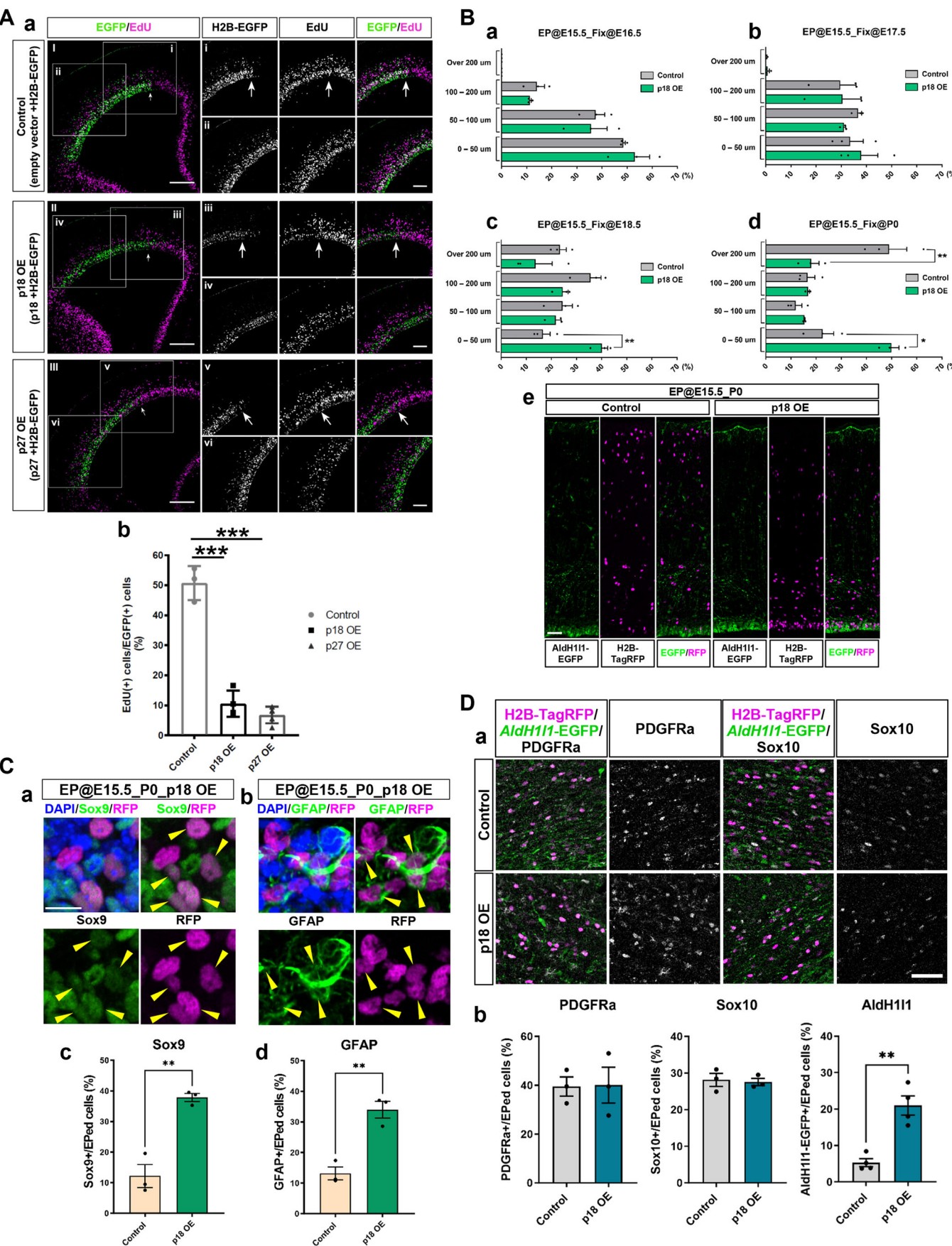

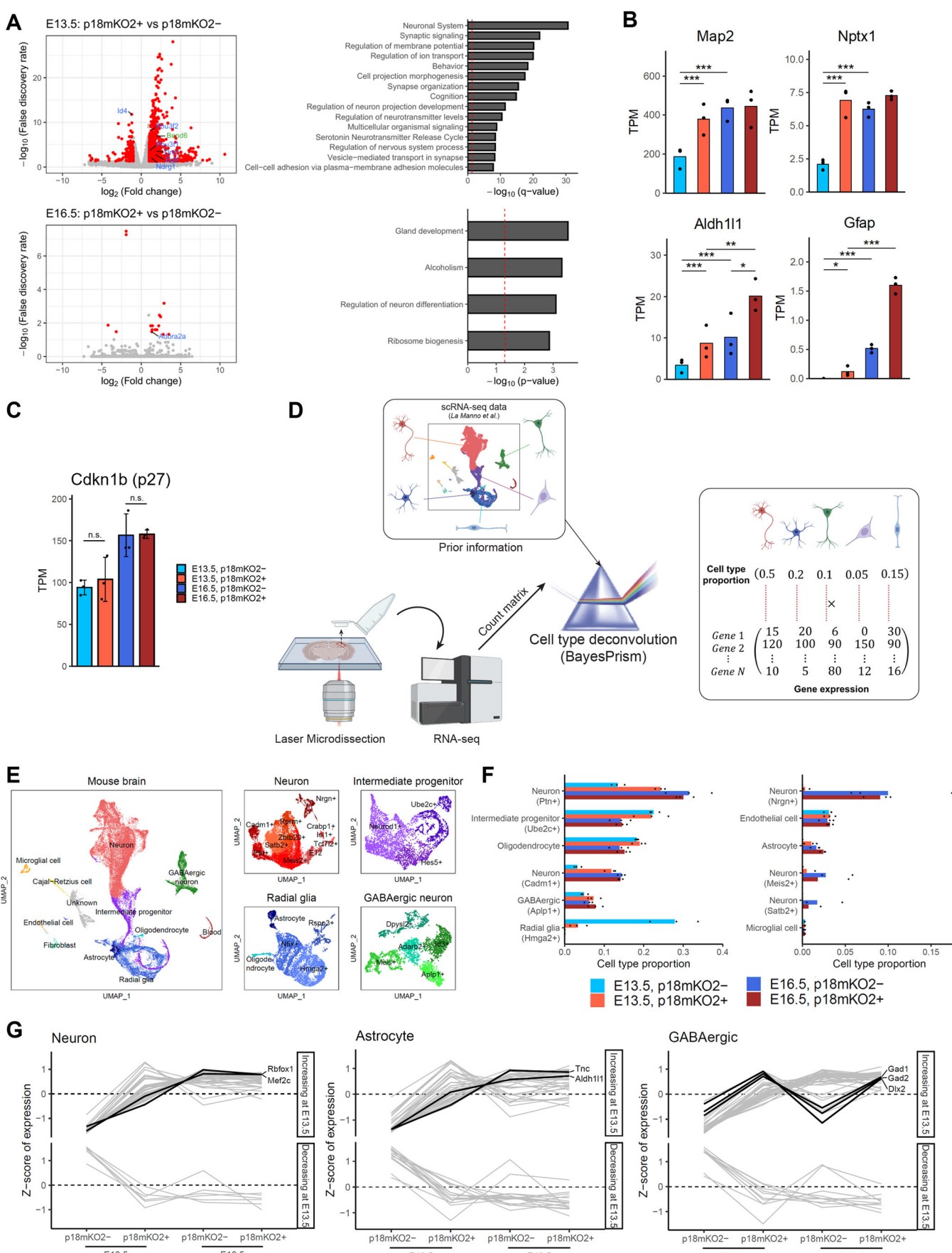

◀ **Figure EV3. Cell type deconvolution by BayesPrism.**

(A) Differential expression analysis comparing mKO2- (Control) and mKO2+ (p18 OE) at each stage. At E13.5 and E16.5, 1,211 and 273 DEGs were identified, respectively. There were changes in the expression levels of genes related to "regulation of neural differentiation" at E13.5, whereas no terms related to brain development were found at E16.5. (B) The expression levels of neural markers (*Map2* and *Nptx1*) and astrocyte markers (*Aldh1l1* and *Gfap*) in mKO2- and mKO2+ mice at E13.5 and E16.5. FDR was calculated by DESeq2. From top to bottom, left to right, ***$P = = 1.44e\text{-}11$, ***$P = 7.15e\text{-}7$ (Map2), ***$P = 3.63e\text{-}11$, ***$P = 1.28e\text{-}10$ (Nptx1), **$P = 0.000234$, ***$P = 1.05e\text{-}9$, ***$P = 6.14e\text{-}7$, *$P = 0.00120$ (AldH1l1), ***$P = 1.36e\text{-}5$, ***$P = 1.58e\text{-}5$, *$P = 0.00936$ (Gfap). (C) The expression levels of *p27* (*Cdkn1b*) in mKO2- and mKO2+ mice at E13.5 and E16.5 ($n = 3$ for each sample). FDR was calculated by DESeq2; error bars show mean ± SD. No statistically significant differences were detected. (D) Schema of the analysis pipeline for the cell type deconvolution. (E) UMAPs of scRNA-seq from the developing mouse brain (La Manno et al, 2021). Subtypes of Neuron (*Mapt +*, *Ly6h +*), Intermediate progenitor (*Eomes +*), Radial glia (*Nes +*, *Vim +*) and GABAergic neuron (*Gad1 +*, *Gad2 +*) were indicated by their marker genes. (F) Full deconvolution result using BayesPrism. Each dot represents a replicate of each sample. (G) The expression level of the top 50 variable genes found in Fig. 6F at E13.5 and E16.5. Representative genes are indicated by a thick black line.

**A**

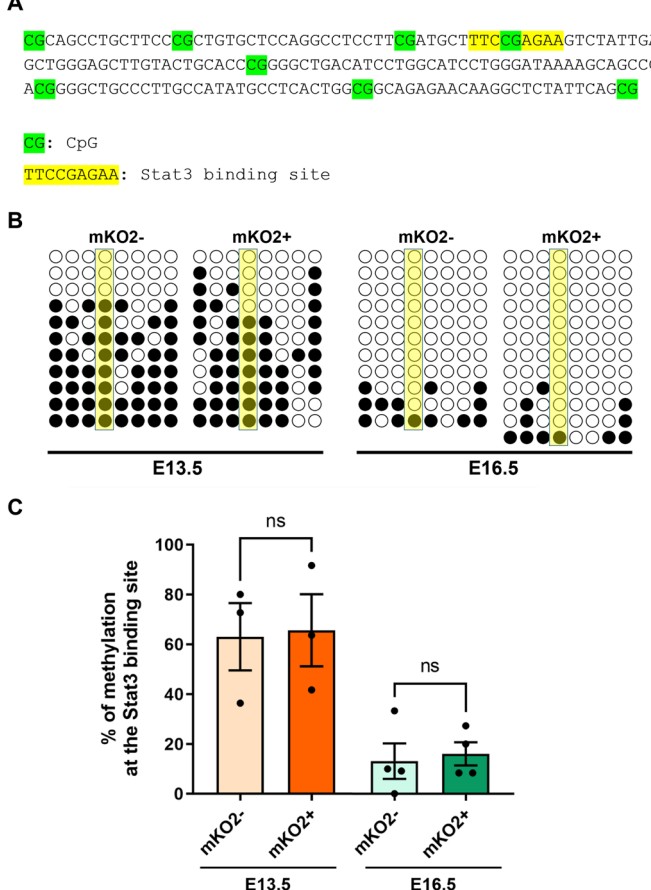

CGCAGCCTGCTTCCCGCTGTGCTCCAGGCCTCCTTCGATGCTTTCCGAGAAGTCTATTGA
GCTGGGAGCTTGTACTGCACCCGGGGCTGACATCCTGGCATCCTGGGATAAAAGCAGCCC
ACGGGGCTGCCCTTGCCATATGCCTCACTGGCGGCAGAGAACAAGGCTCTATTCAGCG

CG: CpG

TTCCGAGAA: Stat3 binding site

**B**

**C**

**Figure EV4. Bisulfite sequencing analysis to investigate the DNA methylation status.**

(**A**) Positions of CpG (green) and the Stat3 binding site at the promoter region of GFAP (yellow) investigated in a previous study (Takizawa et al, 2001). (**B**) Representative results showing methylated sites (black) among CpG positions (circle) in the cortical VZ of p18-P2A-mKO2-negative or -positive mice. Tamoxifen was administered to pregnant R26R-p18-P2A-mKO2/Nes-CreER^T2 mice at E12.5 and 15.5, and then embryonic brains were dissected at E13.5 and 16.5. After the preparation of fresh brain sections, the genomic DNA from the VZ isolated by LMD was prepared for bisulfite assay. The yellow color indicates the Stat3 binding site. (**C**) Quantification of the methylation status at the Stat3 binding site at E13.5 and E16.5 in mKO2- and mKO2+ mice ($n = 3$ for E13.5 and 4 for E16.5 brains). Two-tailed $t$-test; error bars show mean ± SEM. No statistically significant differences were detected.

