## [Peer Review File · The EMBO Journal]

Cyclin-dependent kinase inhibitor p18 regulates lineage transitions of excitatory neurons, astrocytes, and interneurons in the mouse cortex

Yoichi Kosodo, Wonyoung Lee, Byunghee Kang, Hyo-Min Kim, Tsuyoshi Ishida, Minkyung Shin, Misato Iwashita, Masahiro Nitta, Aki Shiraishi, Hiroshi Kiyonari, Koichiro Shimoya, Kazuto Masamoto, and Tae-Young Roh

Corresponding author(s): Yoichi Kosodo (kosodo@kbri.re.kr) , Tae-Young Roh (tyroh@ewha.ac.kr)

Review Timeline:

Transfer Date:	30th Jul 24
Editorial Decision:	20th Sep 24
Revision Received:	30th Sep 24
Editorial Decision:	4th Nov 24
Revision Received:	13th Nov 24
Accepted:	14th Nov 24

Editor: Ioannis Papaioannou

Transaction Report: Please note that the manuscript was transferred from another journal where it was originally reviewed.

Authors' appeal letter to previous reviewers' comments:

Reviewer #1 (Remarks to the Author):

The authors have answered to my initial points of concern at full satisfaction, hence I endorse publication of this very nice, elegant and interesting study.

Reviewer #3 (Remarks to the Author):

Unfortunately, this reviewer believes that the authors have not sufficiently addressed the proposed comments to clarify the main pitfalls of this study. Despite the notes incorporated, it remains unclear whether p18 and Dlx2 play a fundamental role in the fate switch from neurogenesis to gliogenesis and interneuron production during embryonic neural development.

1) The authors emphasize a key experiment in which they lineage trace radial glia cells and assess the progeny these cells generate upon p18 overexpression (OE). They label progenitor cells with a PB-RFP construct while introducing a stable reporter (PB-Gfa2-EGFP) to label astrocytes originating from these progenitors. The authors equate the transient expression of EGFP and p18, assuming all to be the same astrocytes, and claim that Dlx2-positive cells arise from these EGFP+ cells. However, they fail to incorporate a stably-expressed fluorescent reporter for p18 and/or Dlx2 expression, which would be crucial for accurately tracking the progeny of p18+ or Dlx2+ cells and determining how changes in p18 or Dlx2 expression affect astrocyte and olfactory bulb (OB) interneuron production.

Answer:

The original Reviewer #3's point was the following. *"It would be beneficial to provide evidence of the expression of these genes in relation to one another over time, perhaps through immunostaining or reporters. Ideally, information of sequential expression at the single cell level would clarify if there is a switch from neurogenesis to gliogenesis and back to neurogenesis within the same progenitor or rather a sequential activation of different subset of progenitors. (quote from the previous Reviewers' comments)". Here, employing immunostaining was indicated as one of the options (1st underlined point of the sentence above). Also, there were substantial concerns about establishing accurate reporter lines*

(described in the next paragraph). Therefore, we chose the clonal analysis using immunostaining of Dlx2 and the reporter EGFP expression to recognize the footprint of astrocyte lineage, then showed the transition of the lineage in control (considered as endogenous), p18 OE, and the knockdown of p18 and p27 cases in several developmental stages at the single cell resolution (2nd underlined point). However, Reviewer #3 now insists in the decision letter that making a stably-expressed fluorescent reporter for p18 and/or Dlx2 expression is indispensable. Therefore, I request the rationality of our strategy in terms of the original scope of the study to be judged by cross-reviewing.

Below, I describe why we did not utilize “a stably-expressed fluorescent reporter for p18 and/or Dlx2 expression.” In short, the accuracy of the strategy is highly suspectable because of the following reasons.

Many studies have reported the multifaced regulatory mechanisms of p18 expression. These include hypermethylation in the promoter region (Sanchez-Aguilera et al., 2004) and the synergistic enhancement of the promoter by two transiently expressed transcription factors (E2F1 and SP1) during the cell cycle (Blais et al., 2002). Furthermore, p18 transcripts can be generated by promoter switching (Phelps et al., 1998), and the p18 gene contains two distinct promoters that are activated differently depending on the differentiation status of the cells (Pankratova, 2008). The complexity of these properties of p18 regulatory mechanisms poses a challenge in generating a stable and reliable reporter for p18, which accurately reflects endogenous p18 expression. Furthermore, our knockdown analysis shows that double-knockdown of p18 and p27 reduces astrocyte production compared to the single knockdown (Main Figure 1 and 2C, Supplemental Figure 9). Therefore, it is not enough to generate a single p18-reporter, but p18 and p27 dual-reporters are required to observe the function and dynamics of CDKIs related to the lineage transition.

The expression of Dlx2 is also subjected to several regulatory mechanisms. RNA and protein expression of the Dlx transcription factors (TFs) (Dlx2/5/1 and 6) is initiated at E9.5 and follows a temporal expression regulatory system based on the developmental stages (Eisenstat et al., 1999). Their expression is shown in an overlapping region, suggesting that the different Dlx TFs compensate for each other. Indeed, there are many studies suggesting that Dlx-family genes perform dual activating and repressing functions and share their binding properties (Panganiban and Rubenstein, 2002) (Lindtner et al., 2019). Consistent with the view, our RNA-seq results demonstrate that Dlx2/5/1 and 6 were all upregulated in the knock-in mouse at E16.5 after inducing p18 OE at E15.5 (Main Figure 6F).

Given the high-order regulatory mechanisms of p18/Dlx2 expression, it is a valid concern whether the transcripts and proteins with fluorescent reporters behave precisely the same as endogenous products. Furthermore, it is reported that CDKIs are subject to ubiquitination during the cell cycle (Forget et al., 2008). Therefore, the protein degradation of fluorescent reporters also needs to be controlled in the same way as endogenous p18 and p27 to address their dynamics. For these reasons, we employed the clonal analysis using an alternative reporter system with immunostaining to address Reviewer #3's point.

For this purpose, the Piggy Bac (PB) reporter system combined with *in utero* EP (IUE) is ideal. Chromosome-integrated PB-reporters introduced by IUE allow the labeling of all clones derived from electroporated dorsal apical progenitors. We utilized the plasmids of PB-RFP and PB-*Gfa2*-EGFP, which is already well characterized as a reporter of astrocyte lineage (Hamabe-Horiike et al., 2021), at the time of electroporation. Essentially all progenies from the electroporated cells are labeled by PB-RFP. Once astrocyte lineage is induced, the cell becomes EGFP-positive, then the EGFP signal remains for a certain period. After the fixation, we stained the brain section using an anti-Dlx2 antibody to detect the endogenous Dlx2 protein. Several studies already clarified the function of the Dlx-family gene to induce the interneuron lineage. In the mouse brain, the lineage transition from dorsal cortical progenitors to OB interneuron via Dlx2 has already been demonstrated by studies using the *Emx1* reporter mouse (Kohwi et al., 2007) and the single-cell RNAseq (Li et al., 2021). Therefore, the immunoreactivity of Dlx2 in *Gfa2*-EGFP-positive cells indicates that the lineage transition from astrocyte to OB interneuron has taken place (Rebuttal Figure 1, Main Figure 9D and 9E).

Rebuttal Figure 1. Investigation of the stage-dependent lineage transitions from astrocyte to interneuron at the single-cell level

Consistent with other experiments in Figures 7 and 8, around 5% of progenies of electroporated cells were Dlx2-positive at the perinatal stage in the control (Rebuttal Figure 1, Main Figure 9D and 9E). Among PB-RFP/Dlx2 double-positive cells, 25 % of cells were *Gfa2*-EGFP positive (yellow arrowheads) at the P1 stage (blue arrows, after IUE at E15.5). Notably, this ratio increased to 74 % due to the enhanced expression of p18 at the same stage (red arrows). By knocking down endogenous p18 and p27, we observed decreased Dlx2 positive cells at E17.5, while no Dlx2 positive cells with astrocyte footprint were identified at P1 and P4 after IUE at E15.5 (Main Figure 9D and 9E). Thus, the stage-dependent lineage transition from astrocyte to OB interneuron was verified at the single-cell resolution by the clonal analysis using *Gfa2*-EGFP reporter and Dlx2 immunostaining, and the potency of p18 to facilitate the lineage transition was demonstrated.

In summary, the primary focus of our study, that is, the function of CDKIs to regulate the lineage border of neural stem cells' progeny, has been adequately addressed at the single-cell resolution *in vivo*. We acknowledge the viewpoint that investigating the dynamic behavior of endogenous molecules can provide further insights. The use of live imaging could potentially uncover the time course during the lineage transition. Moreover, to gain a comprehensive understanding of the dynamics of endogenous CDKIs, it is crucial to monitor not only the gene expression but also the protein degradation, as they are subject to ubiquitination during the cell cycle as described. We include this argument as a limitation of the study in the Discussion in the newly revised manuscript (track-changed).

2) The authors also include an interesting experiment where they mis-express p18 at an earlier timepoint (E12.5) and assess changes in the progeny produced (Figure S4). Progeny production was studied at P10, where no differences were found in the proportions of astrocytes or neurons generated. These results add to the uncertainty about p18's essential role in the glial switch during brain development. It would have been beneficial to assess the effects of p18 mis-expression at earlier timepoints to determine the immediate targets of p18.

Along these lines, the authors should further explore the possibility of p18 OE inducing cell cycle exit and immediate cell differentiation in future analyses.

Answer:

The p18 OE at the earlier time point was meticulously tested in plasmid-based *in utero* electroporation and Cre-loxP recombination in a p18-mKO2 knock-in mouse.

We first observed the upregulation of neurogenin2 (Ngn2) positive cells, a well-known transcription factor to induce excitatory neurons, one day after the introduction of p18 and p27 by *in utero* EP at E12.5 (Blue rectangle in Rebuttal Figure 2A, Main Figures 3A and 3B). These results suggest that the CDKIs enhance the neural induction at the stage, as reported (Nguyen et al., 2006). We investigated whether there are differences between the ratio of deeper (Ctip2+) and upper (Satb2+) layer neurons, populations generated sequentially in the dorsal cortex. We did not see significant differences between the control and P18 OE, indicating that the regulation of neural subtypes may not be relevant to CDKIs. Using the same condition, we quantified the ratio of deeper neurons and astrocytes (Rebuttal Figure 2B, Main Figure 3D, Supplemental Figures 4A and 4B). Again, there are no significant changes.

The situation took a fascinating turn when we performed *in utero* EP at E15.5, the stage when Satb2+ upper layer neuron is dominantly produced. After one day, we observed the induction of Aldh1l1+ and the reduction of Ngn2+ cells (Red rectangle in Rebuttal Figure 2A, Main Figures 2F, 2G, 3A, and 3B). At the P10 stage, we found that Satb2- and Cux1-positive upper-layer neurons decreased while GFAP- and S100beta-positive astrocytes increased (Rebuttal Figure 2C, Main Figure 3E). The results were further confirmed using the Nestin-CreERT2/p18-mKO2 knock-in mice by applying tamoxifen at E15.5 to induce excess p18 in the neural stem cells, followed by the analysis at the P10 stage (Rebuttal Figure 2D, Main Figures 5D and 5E). In contrast, the double knock-down of p18 and p27 increased upper-layer neurons while decreasing astrocytes observed both in mRNA (Supplemental Figure 9) and in protein (Main Figure 9A-C) levels.

Rebuttal Figure 2. Experimental results documenting the stage-dependent roles of p18

These results imply that the p18 OE enhances neurogenesis at the earlier stage while inducing the lineage switch from the upper layer neuron to the astrocyte at the later stage of cortical development. The stage-dependent roles of p18 were also confirmed by RNAseq using knock-in mice. Gene ontology and deconvolution analysis revealed the immediately responded biological network and gene sets by the p18 induction in each stage (Main Figure 6 and Supplemental Figure 6; details are described in Section 3 of this letter).

Additionally, we have confirmed that the cell cycle exits by p18 OE (Supplemental Figure 3A).

3) Additionally, the authors have not adequately analyzed their RNAseq data. It is strongly recommended to use a linear model to identify transcriptional changes correlated with timepoint and genotype to better understand the effects of p18 OE during neural development.

Answer:

The linear model mentioned by the reviewer involves comparing Control and OE at each stage, which is a common approach in overexpression experiments. At E13.5 and E16.5, we identified 1,211 and 273 differentially expressed genes (DEGs), respectively (Rebuttal Figure 3A). At E13.5, there were changes in the expression levels of genes related to "regulation of neural differentiation," whereas no terms related to brain development were found at E16.5 (Rebuttal Figure 3B). Additionally, terms related to glial cell differentiation did not appear at either stage.

(A) # of DEGs
Threshold: FDR < 0.05, FC > 2

Contrast	Up-regulated	Down-regulated
E13.5, p18mKO+ vs p18mKO-	929	282
E16.5, p18mKO+ vs p18mKO-	90	183

(B) E13.5, p18mKO+ vs p18mKO-

E16.5, p18mKO+ vs p18mKO-

Rebuttal Figure 3. Differential expression analysis comparing Control and OE at each stage. (A) The number of differentially expressed genes for each comparison. **(B)** Functional enrichment analysis results using DEGs.

Our comparison method involved comparing E13.5 and E16.5 within each condition and then comparing the terms identified in each condition. In Control and OE, we found 2,786 and 701 DEGs, respectively (Rebuttal Figure 4A, Main Figure 6A and 6B). In Control, terms such as "mitotic cell cycle process" and "synapse organization" were identified, while in OE, terms related to "regulation of neuron differentiation" and "glial cell differentiation" were found (Red arrows in Rebuttal Figure 4B, Main Figure 6C and 6D).

(A) # of DEGs
Threshold: FDR < 0.05, FC > 2

Contrast	Up-regulated	Down-regulated
p18mKO-, E16.5 vs E13.5	1,780	1,006
p18mKO+, E16.5 vs E13.5	377	324

(B) p18mKO-, E16.5 vs E13.5

p18mKO+, E16.5 vs E13.5

Rebuttal Figure 4. Differential expression analysis comparing E13.5 and E16.5 for each condition. (A) The number of differentially expressed genes for each comparison. **(B)** Functional enrichment analysis results using DEGs.

Deconvolution analysis showed an increase in the proportion of astrocytes with p18 OE at both E13.5 and E16.5 (Rebuttal Figure 5, Main Figure 6E). Importantly, the proportion of increased Astrocyte was minor compared to the increased proportion of Neuron in E13.5, while only Astrocyte was increased in E16.5, but not Neuron, Radial glia, Intermediate progenitor, and Oligodendrocyte. Additionally, the proportion of astrocytes increased in both Control and OE, indicating that glial differentiation occurs with stage progression regardless of p18 OE.

Rebuttal Figure 5. Proportion of each cell in each sample after RNA-seq deconvolution analysis.

Since glial cell differentiation occurs in the Control condition as well, comparing Control and OE at each stage does not adequately explain the increase in astrocyte proportion. Moreover, by first comparing stages within each condition and then comparing the terms identified in Control and OE, we can better understand biological processes related to the differentiation by p18 OE. Consequently, we conducted functional enrichment analyses within each condition to identify key biological process terms that show major differences.

In summary, despite the authors' efforts to address the proposed claims, the main question of the study remains unresolved: there is insufficient evidence to assert that p18 regulates the lineage switch from neurogenesis to gliogenesis and further induces OB interneuron production through *Dlx2* expression.

Answer:

The main questions and novelties we have revealed in this study are the following:

1. Gene knock-down analysis revealed that endogenous p18, together with p27, is required for neural and glial lineage transitions at the proper timing.
2. Augmented p18 enhances neural production at the earlier stages, while the lineage induction of astrocytes at the later stages of embryonic brain development.
3. At the perinatal stage, p18 potentiates the induction OB interneurons via the up-regulation of *Dlx2*.

4. The clonal analysis using the PB-*Gfa2*-EGFP reporter documented the transitions from the astrocyte lineage to the *Dlx2*-positive cells at the single level, regulated by the function of CDKIs.

Thus, our results demonstrate the necessity and potency of CDKIs in sequentially determining the boundaries among different cellular lineages arising from NSCs in the dorsal cortex.

References in the letter

- Blais, A., D. Monte, F. Pouliot, and C. Labrie. 2002. Regulation of the human cyclin-dependent kinase inhibitor p18INK4c by the transcription factors E2F1 and Sp1. *J Biol Chem.* 277:31679-31693.
- Eisenstat, D.D., J.K. Liu, M. Mione, W. Zhong, G. Yu, S.A. Anderson, I. Ghattas, L. Puelles, and J.L. Rubenstein. 1999. DLX-1, DLX-2, and DLX-5 expression define distinct stages of basal forebrain differentiation. *J Comp Neurol.* 414:217-237.
- Forget, A., O. Ayrault, W. den Besten, M.L. Kuo, C.J. Sherr, and M.F. Roussel. 2008. Differential post-transcriptional regulation of two Ink4 proteins, p18 Ink4c and p19 Ink4d. *Cell Cycle.* 7:3737-3746.
- Hamabe-Horiike, T., K. Kawasaki, M. Sakashita, C. Ishizu, T. Yoshizaki, S.I. Harada, K. Ogawa-Ochiai, Y. Shinmyo, and H. Kawasaki. 2021. Glial cell type-specific gene expression in the mouse cerebrum using the piggyBac system and in utero electroporation. *Sci Rep.* 11:4864.
- Kohwi, M., M.A. Petryniak, J.E. Long, M. Ekker, K. Obata, Y. Yanagawa, J.L. Rubenstein, and A. Alvarez-Buylla. 2007. A subpopulation of olfactory bulb GABAergic interneurons is derived from *Emx1*- and *Dlx5/6*-expressing progenitors. *J Neurosci.* 27:6878-6891.
- Li, X., G. Liu, L. Yang, Z. Li, Z. Zhang, Z. Xu, Y. Cai, H. Du, Z. Su, Z. Wang, Y. Duan, H. Chen, Z. Shang, Y. You, Q. Zhang, M. He, B. Chen, and Z. Yang. 2021. Decoding Cortical Glial Cell Development. *Neurosci Bull.* 37:440-460.
- Lindtner, S., R. Catta-Preta, H. Tian, L. Su-Feher, J.D. Price, D.E. Dickel, V. Greiner, S.N. Silberberg, G.L. McKinsey, M.T. McManus, L.A. Pennacchio, A. Visel, A.S. Nord, and J.L.R. Rubenstein. 2019. Genomic Resolution of DLX-Orchestrated

- Transcriptional Circuits Driving Development of Forebrain GABAergic Neurons. *Cell Rep.* 28:2048-2063 e2048.
- Nguyen, L., A. Besson, J.I. Heng, C. Schuurmans, L. Teboul, C. Parras, A. Philpott, J.M. Roberts, and F. Guillemot. 2006. p27kip1 independently promotes neuronal differentiation and migration in the cerebral cortex. *Genes Dev.* 20:1511-1524.
- Panganiban, G., and J.L. Rubenstein. 2002. Developmental functions of the Distal-less/Dlx homeobox genes. *Development.* 129:4371-4386.
- Pankratova, E.V. 2008. [Alternative promoters and the complexity of the mammalian transcriptome]. *Mol Biol (Mosk).* 42:422-433.
- Phelps, D.E., K.M. Hsiao, Y. Li, N. Hu, D.S. Franklin, E. Westphal, E.Y. Lee, and Y. Xiong. 1998. Coupled transcriptional and translational control of cyclin-dependent kinase inhibitor p18INK4c expression during myogenesis. *Mol Cell Biol.* 18:2334-2343.
- Sanchez-Aguilera, A., J. Delgado, F.I. Camacho, M. Sanchez-Beato, L. Sanchez, C. Montalban, M.F. Fresno, C. Martin, M.A. Piris, and J.F. Garcia. 2004. Silencing of the p18INK4c gene by promoter hypermethylation in Reed-Sternberg cells in Hodgkin lymphomas. *Blood.* 103:2351-2357.

Dear Yoichi,

Thank you again for transferring to The EMBO Journal for our consideration your manuscript (EMBOJ-2024-118610-T) along with the review history and your rebuttal letter to the comments of the reviewers who previously assessed it at another journal. We have shared your manuscript and your rebuttal letter with an additional arbitrator with familiarity both with the field and our journal and its scope, and we have now received their comments, which I have already shared with you (they are included again below). Thank you again for your patience during this rather protracted process.

I am glad to say that our arbitrator is generally supportive of the manuscript, rates the overall conceptual novelty and significance of the study highly, and finds the majority of the previously raised concerns adequately and sufficiently addressed in your revised manuscript. There are only two remaining suggestions that should be addressed for the manuscript to be accepted for publication in The EMBO Journal:

1. Regarding the first main concern of reviewer 3: "... the authors should show that p18OE actually promotes OB neuron production (%Sp8+/RFP+) at later stages in this model (PB vectors-Fig9)."
2. Regarding the third main concern of reviewer 3: "... the authors show the results of "linear" analysis (Rebuttal Fig 3), including a volcano plot analysis (Sup.Fig)".

In addition, the arbitrator points out that the multitude of experimental approaches used in your study complicates integration of the results, and considering also the broad readership of our journal, we would encourage you to take on board the arbitrator's advice that the text would benefit from better clarity, and that the title needs to be revised.

In light of the review history of your manuscript and the favorable input we received from our additional arbitrator, I would like to invite you to submit a final version of your manuscript addressing the remaining points mentioned above, along with a detailed description of any changes and additions made to the manuscript. Please let me know if you have any questions or comments that you would like to discuss with me.

As a matter of policy, competing manuscripts published during the revision period will not negatively impact our assessment of the conceptual advance presented by your study. However, we request that you contact us as soon as possible upon publication of any related work, to discuss how to proceed.

Thank you again for the opportunity to consider your work for publication in The EMBO Journal. I am looking forward to the revised version of your manuscript.

Best regards,

Ioannis

Instructions for preparing your revised manuscript

1. When you are ready to submit the revision, please upload:

- A Word file of the manuscript text (including legends of main Figures, EV Figures and Tables). Please make sure that changes are highlighted (or "tracked") to be clearly visible.

- Individual production-quality figure files (one file per figure). When assembling your figures, please refer to our figure preparation guidelines in order to ensure proper formatting and readability in print as well as on screen:

If the data shown in a figure are obtained from n {less than or equal to} 2, please use scatter plots showing the individual data points.

- i. the name of the statistical test used to generate error bars and P values
- ii. the number (n) of independent experiments (please specify technical or biological replicates) underlying each data point (discussion of statistical methodology can be reported in the Materials and Methods section, but figure legends should contain a basic description of n, P, and the test applied)
- iii. the nature of the bars and error bars (s.d., s.e.m.).

- A point-by-point response to the referees' comments, with a detailed description of the changes made (as a word file). All referees' concerns must be fully addressed and their suggestions taken on board. When preparing your letter of response to the referees' comments, please bear in mind that this will form part of the Review Process File and will therefore be available online to the community. Please note that you have the possibility to opt out of the transparent process at any stage prior to publication by letting the editorial office know (contact@embojournal.org); if you do opt out, the Review Process File link will point to the following statement: "No Peer Review File is available with this article, as the authors have chosen not to make the review process public in this case.". For more details on our Transparent Editorial Process, please visit our website: <https://www.embopress.org/page/journal/14602075/authorguide#transparentprocess>

- Expanded View (EV) files (replacing Supplementary Information) that are collapsible/expandable online. A maximum of 5 EV Figures can be typeset. EV Figures should be cited as "Figure EV1, Figure EV2" etc. in the text, and their respective legends should be included in the manuscript file after the legends of regular figures. See detailed instructions regarding Expanded View files here:

- For the figures that you do NOT wish to display as Expanded View figures, they should be bundled together with their legends in a single PDF file called "Appendix", which should start with a short Table of Contents (including page numbers). Appendix figures should be referred to in the main text as: "Appendix Figure S1, Appendix Figure S2" etc. Please see detailed instructions here: <https://www.embopress.org/page/journal/14602075/authorguide#expandedview>

- A complete author checklist, which you can download from our author guidelines (<https://www.embopress.org/page/journal/14602075/authorguide>). Please note that the checklist will also be part of the Review Process File.

2. Please note that no statistics should be calculated and shown in Figures if n=2. Please also note that each p value should be reported as an exact value.

3. Before submitting your revision, primary datasets (and computer code, where appropriate) produced in this study need to be deposited in appropriate public databases (see <https://www.embopress.org/page/journal/14602075/authorguide#dataavailability>). Their accession numbers, databases, and the specific URLs (links) should be listed in a formal "Data availability" section (placed after Methods). In particular, we kindly request that the RNAseq data produced in this study are deposited to an appropriate repository.

*** The Data Availability Section is restricted to new primary data that are part of this study. In case you have no data that require deposition in a public database, please state so instead of referring to the database: "Our study includes no data deposited in public repositories." under the heading "Data availability". ***

*** All links should resolve to a page where the data can be accessed. ***

*** Please remember to provide in the Data availability section of your revised manuscript reviewer passwords if the datasets are not yet public. ***

*** Please use detailed data citations for already available datasets that were re-analyzed in your study - for more information on the format, see point #9 below. ***

4. Please check that the title and the abstract of the manuscript are brief, yet explicit, even to non-specialists. The length of the title should not exceed 100 characters, and the abstract should be a single paragraph not exceeding 175 words.

5. All materials and methods need to be described in the manuscript using our "Structured Methods" format, which is now required for all research articles. According to this format, the Methods section includes a single "Reagents and Tools Table" - listing key reagents, experimental models, software and relevant equipment including their sources and relevant identifiers- followed by a "Methods and Protocols" section describing the methods. Please download and fill our Reagents and Tools Table template (.docx), which you can find in our author guide:

<https://www.embopress.org/page/journal/14602075/authorguide#structuredmethods>. When submitting your revised manuscript, please do not include the Reagents and Tools Table in the Methods section of the manuscript but upload it as a separate file choosing the file type "Reagent Table".

6. Please also note our reference format: <https://www.embopress.org/page/journal/14602075/authorguide#referencesformat>.
7. At EMBO Press we ask authors to provide source data for the main manuscript figures. Our source data coordinator will contact you to discuss which figure panels we would need source data for and will also provide you with helpful tips on how to upload and organize the files.
8. Please remember: digital image enhancement is acceptable practice, as long as it accurately represents the original data and conforms to community standards. If a figure has been subjected to significant electronic manipulation, this must be noted in the figure legend or in the "Materials and Methods" section. The editors reserve the right to request original versions of figures and the original images that were used to assemble the figure.
9. Our journal encourages inclusion of data citations in the reference list to directly cite datasets that were obtained from public databases. Data citations in the article text are distinct from normal bibliographical citations and should directly link to the database records from which the data can be accessed. In the main text, data citations are formatted as follows: "Data ref: Smith et al, 2001" or "Data ref: NCBI Sequence Read Archive PRJNA342805, 2017". In the Reference list, data citations must be labeled with "[DATASET]". A data reference must provide the database name, accession number/identifiers, and a resolvable link to the landing page from which the data can be accessed at the end of the reference. Further instructions are available at: <https://www.embopress.org/page/journal/14602075/authorguide#referencesformat>.
10. We request authors to consider both actual and perceived competing interests. Please review our policy (<https://www.embopress.org/page/journal/14602075/authorguide#conflictsofinterest>) and update your competing interests statement if necessary. Please name this section 'Disclosure and competing interests statement' and place it after the Acknowledgements section.
11. Please note that all corresponding authors are required to provide an ORCID ID upon submission of a revised manuscript (<https://orcid.org/>). Please find instructions on how to link your ORCID ID to your account in our manuscript tracking system in our Author guidelines (<https://www.embopress.org/page/journal/14602075/authorguide#authorshipguidelines>).
12. We use CRediT to specify the contributions of each author in the journal submission system. CRediT replaces the author contribution section, which should be removed from the manuscript. Please use the free text box to provide more detailed descriptions. See also guide to authors: <https://www.embopress.org/page/journal/14602075/authorguide#authorshipguidelines>.
13. Further information is available in our Guide For Authors: <https://www.embopress.org/page/journal/14602075/authorguide>
14. We would also welcome the submission of cover suggestions or motifs to be used by our Graphics Illustrator in designing a cover.

Arbitrator #1:

In this study, the authors study the role of p18 and p27, two cyclin-dependent inhibitors, during mouse cortex development, by modulating their expression levels in different mouse models.

Using a reporter mouse line that expresses GFP under the control of the *Aldh1l1* locus, a marker of astrocytes, they show that the expression levels of p18 and p27 gradually increase during development, with a peak of p18 at E15.5, at the onset of astrocyte generation.

Using in utero electroporation, they then show that p18 and p27 simple or double knock-down (KD) blocked astrocytic differentiation.

They then focus on p18 and show that the effect of p18 OE is stage-dependent: p18 OE at early stage (E12.5) causes neuronal differentiation with increased *Ngn2* expression, while at later stage (E15.5) it drives astrocytic differentiation at later stages (E16.5 towards P0), and suppresses neuronal differentiation through *dlx2*-mediated *Ngn2* repression.

They then used a conditional p18-P2A-mKO2 knock-in mouse line, that enabled them to confirm increased astrocytic differentiation and decreased neuronal differentiation upon p18OE at late stages (E15.5). Transcriptomic analyses after short-term p18OE to detect immediate response genes, point in the same direction: increased expression of neuronal markers upon p18OE at early developmental stage vs late stage, and increased expression of glial markers upon p18OE at late stage.

This transcriptomic analysis leads them to *Dlx2*, which is upregulated in p18OE cells. *Dlx2* OE at late stage (E15,5) inhibits neuronal and astrocytic differentiation, while *Dlx2* KD leads to opposite effects. P18 induces *Dlx2* expression that subsequently promotes OB interneuron differentiation at late stages.

To trace these lineage transitions (NSCs>astrocytes>OB neurons) at the single-cell level, they used a transposon-based fluorescent reporter of astrocytic lineage.

They conclude that p18 influences the fate switch from neurogenesis to gliogenesis and to interneuron production during mouse cortical development.

We have 3 major comments :

-The authors used a significant number of different approaches and mouse models, which sometimes makes it difficult to integrate all the results.

- Overall, the text would benefit from greater clarity.

- The title should be changed in order to focus on p18 rather than CDKi and should mention interneurons.

Overall, we think that the authors have correctly addressed the queries of the previous reviewers, which led to significant additions that have substantially improved the ms and strengthened their conclusions.

Below our assessment of R3:

Reviewer #3 raised three main concerns:

#1: whether the lineage transition (cortical neurons>astrocytes>OB interneuron) occurs within a single NSC or distinct subsets.

In the first round of review, R3 suggests to use either immunostaining or ideally reporters to trace the transitions.

To answer this important question, the authors used Piggy-Bac (PB) RFP (PB-RFP) to track the progeny of the cells, and a stably expressed reporter, PB-Gfa2-EGFP, to monitor the induction of the astrocyte lineage. They found that P18OE at E15,5 increased the proportion of GFP+/Dlx2+ among RFP+ cells at perinatal stages (E17,5/P1) (around 7 to 13%?), which supposedly leads to OB neuron production.

The use of fluorescent reporters to track p18 and *Dlx2* expression would in theory allow for more precise tracing of lineages transition (see R3 comment-second review). However, we share the authors' view that the complexity of the regulation of p18 and *Dlx2* expression questions the validity of using such reporters, as requested by R3 in the second round of Review.

Importantly, we think that the authors should show that p18OE actually promotes OB neuron production (%Sp8+/RFP+) at later stages in this model (PB vectors-Fig9).

#2: to assess the effects of p18 mis-expression at earlier timepoints to determine the immediate targets of p18

The authors perform answered this query and performed an experiment using the H2B tag RFP model (and not the Cre-loxP recombination in a p18-mKO2 knock-in mouse as mentioned in the appeal). They observed increased proportion of Ngn2+ cells among RFP+ cells at E13,5 upon p18OE at E12,5.

The immediate effects of p18OE were also approached in the RNA seq experiment (Fig.6A) which highlights DEGs between E12,5 and E13,5 in both control and p18OE conditions. This revealed increased expression of neuronal markers in the p18OE condition.

#3 Analysis of RNAseq data. R3 recommended performing a linear analysis (comparing p18OE to control at each developmental stage E13 and E16), which is a classical type of analysis in cases of overexpression. However, we agree that the authors' type of analysis provides important information, allowing the identification of immediate responses to p18OE.

Nevertheless, we suggest that the authors show the results of "linear" analysis (Rebuttal Fig 3), including a volcano plot analysis (Sup.Fig)

Advisor 1:

In this study, the authors study the role of p18 and p27, two cyclin-dependent inhibitors, during mouse cortex development, by modulating their expression levels in different mouse models.

Using a reporter mouse line that expresses GFP under the control of the *Aldh1l1* locus, a marker of astrocytes, they show that the expression levels of p18 and p27 gradually increase during development, with a peak of p18 at E15.5, at the onset of astrocyte generation.

Using in utero electroporation, they then show that p18 and p27 simple or double knock-down (KD) blocked astrocytic differentiation.

They then focus on p18 and show that the effect of p18 OE is stage-dependent: p18 OE at early stage (E12.5) causes neuronal differentiation with increased *Ngn2* expression, while at later stage (E15.5) it drives astrocytic differentiation at later stages (E16.5 towards P0), and suppresses neuronal differentiation through *dlx2* -mediated *Ngn2* repression.

They then used a conditional p18-P2A-mKO2 knock-in mouse line, that enabled them to confirm increased astrocytic differentiation and decreased neuronal differentiation upon p18OE at late stages (E15.5). Transcriptomic analyses after short-term p18OE to detect immediate response genes, point in the same direction: increased expression of neuronal markers upon p18OE at early developmental stage vs late stage, and increased expression of glial markers upon p18OE at late stage.

This transcriptomic analysis leads them to *Dlx2*, which is upregulated in p18OE cells. *Dlx2* OE at late stage (E15.5) inhibits neuronal and astrocytic differentiation, while *Dlx2* KD leads to opposite effects. P18 induces *Dlx2* expression that subsequently promotes OB interneuron differentiation at late stages.

To trace these lineage transitions (NSCs>astrocytes>OB neurons) at the single-cell level, they used a transposon-based fluorescent reporter of astrocytic lineage.

They conclude that p18 influences the fate switch from neurogenesis to gliogenesis and to interneuron production during mouse cortical development.

We have 3 major comments :

- The authors used a significant number of different approaches and mouse models, which sometimes makes it difficult to integrate all the results.
- Overall, the text would benefit from greater clarity.
- The title should be changed in order to focus on p18 rather than CDKi and should mention interneurons.

Answer:

We have changed the title according to the Advisor's comment as well as to follow the journal's Author Instructions. The Abstract has been modified as well to fit the Instructions. Regarding the manuscript's clarity, we have explained the conceptual novelty of the research outcome in the Discussion part. We also carefully re-examined the use of "CDKIs" and "p18" throughout the manuscript to avoid possible confusion.

Overall, we think that the authors have correctly addressed the queries of the previous reviewers, which led to significant additions that have substantially improved the ms and strengthened their conclusions.

Below our assessment of R3:

Reviewer #3 raised three main concerns:

#1: whether the lineage transition (cortical neurons>astrocytes>OB interneuron) occurs within a single NSC or distinct subsets.

In the first round of review, R3 suggests to use either immunostaining or ideally reporters to trace the transitions.

To answer this important question, the authors used Piggy-Bac (PB) RFP (PB-RFP) to track the progeny of the cells, and a stably expressed reporter, PB-Gfa2-EGFP, to monitor the induction of the astrocyte lineage. They found that P18OE at E15,5 increased the proportion of GFP+/Dlx2+ among RFP+ cells at perinatal stages (E17,5/P1) (around 7 to 13%?), which supposedly leads to OB neuron production.

The use of fluorescent reporters to track p18 and Dlx2 expression would in theory allow for more precise tracing of lineages transition (see R3 comment-second review). However, we share the authors' view that the complexity of the regulation of p18 and Dlx2 expression questions the validity of using such reporters, as requested by R3 in the second round of Review.

Importantly, we think that the authors should show that p18OE actually promotes OB neuron production (%Sp8+/RFP+) at later stages in this model (PB vectors-Fig9).

Answer:

We thank the Advisor for raising the point to investigate the cell fate at the later stage because it is important to confirm whether the *Dlx2*-positive population promoted by p18 OE fully entered the interneuron lineage via astrocyte lineage. We electroporated PB-*Gfa2*-EGFP and PB-RFP at E15.5, then brains were fixed at the P4 stage. Brain sections were stained by Sp8 antibody and the rostral migratory stream (RMS) where functional OB interneurons show the chain migration were imaged. Among RFP-positive cells, Sp8-positive cells were significantly increased by p18 OE (28%) compared to the control (14%). Notably, with p18 OE, Sp8/RFP double-positive cells with EGFP signal were dominant (67%). These results confirm the view that p18 facilitates the lineage transitions from NSC to OB interneuron via astrocyte within a single cell.

We have updated Fig. 9D,E to include the new result.

#2: to assess the effects of p18 mis-expression at earlier timepoints to determine the immediate targets of p18

The authors perform answered this query and performed an experiment using the H2B tag RFP model (and not the Cre-loxP recombination in a p18-mKO2 knock-in mouse as mentioned in the appeal). They observed increased proportion of *Ngn2*⁺ cells among RFP⁺ cells at E13,5 upon p18OE at E12,5.

The immediate effects of p18OE were also approached in the RNA seq experiment (Fig.6A) which highlights DEGs between E12,5 and E13,5 in both control and p18OE conditions. This revealed increased expression of neuronal markers in the p18OE condition.

#3 Analysis of RNAseq data. R3 recommended performing a linear analysis (comparing p18OE to control at each developmental stage E13 and E16), which is a classical type of analysis in cases of overexpression. However, we agree that the authors' type of analysis provides important information, allowing the identification of immediate responses to p18OE.

Nevertheless, we suggest that the authors show the results of "linear" analysis (Rebuttal Fig 3), including a volcano plot analysis (Sup.Fig)

Answer:

According to the Advisor's comment, we have added the volcano plot comparing mKO2- and mKO2⁺ at each stage to the new Fig EV3A.

Dear Yoichi,

Thank you again for submitting your revised manuscript (EMBOJ-2024-118610R) to The EMBO Journal for our consideration. As I have already informed you, the arbitrator who previously assessed the previous version of your manuscript has seen the revision and I am glad to say that they are satisfied with it and acknowledge that their remaining concerns have now been successfully addressed (their comments are included below for your information).

I am thus happy to say that your manuscript has now been in principle accepted for publication in The EMBO Journal. Congratulations on an excellent work and thank you for your contribution to our journal!

There are a few minor formatting/editorial changes that we need from you before we can proceed with typesetting and production of proofs for your approval:

- Please reduce the number of keywords to 5 (you currently list 7).
- Please change the heading "Declaration of interest" to "Disclosure and competing interests statement".
- The author contributions statement should be removed from the manuscript file. Instead, we use CRediT to specify the contributions of each author in the journal submission system. Please feel free to use the free text box to provide more detailed descriptions during submission. See also our guide to authors for more information:
<https://www.embopress.org/page/journal/14602075/authorguide#authorshipguidelines>.
- Figures need to be uploaded as individual, high-resolution Figure files.
- The EV Figure legends should be included in the manuscript file below the main Figure legends.
- Please note that EMBO press papers are accompanied online by:
 - A) a short (2 sentences) summary of the findings and their significance,
 - B) 2-5 short bullet points highlighting the key results, and
 - C) a synopsis image in .jpg or .png format that is exactly 550 pixels wide and 300-600 pixels high (the height is variable). Please note that the text needs to be legible at the final size.Please upload this information along with your revised manuscript (the text for A and B should be provided in a separate Word file).
- During our standard Figure checks, we detected two cases of cell re-use in your Figures that are not mentioned in the Figure legends. In particular, we noticed:
 1. Cell re-use between Figure 2F & Figure EV2 E
 2. Cell re-use between Figure 5 (d&b)We kindly ask you to check the content of these Figures and clarify whether cell re-use is intentional and justified by the experimental setup. If this is the case, please note that re-use must be explicitly stated in the respective Figure legends.
- During our routine pre-acceptance checks, our data editors have raised the following queries regarding figures, data, and legends. We acknowledge that some of the following requests have already been addressed -but in some cases not completely, i.e. in only some of the mentioned Figure panels/legends- and we thus kindly ask you to go through the following complete list again and make sure that all requests below are completely addressed in the final version of your manuscript:
 1. Please note that information related to "n" is missing in the legends of Figures 4c-d; 6a-b; EV 3c.
 2. Please note that the error bars are not defined in the legends of Figures 1c, e; 2c, e, g, i; 3b, d, e (a, c, e, g); 4c-d; 5c (a-b), e(a-c); 7b (a-c), e-f, h; 8b, c(c), d(d), f, h; 9b, e; EV 1d (j); EV 2a(b), b(a-d), c(c-d), d(b); EV 3c; EV 4c.
 3. Please note that for heatmap present in Figure 6f; a numbered scale bar is not provided. This needs to be rectified.
 4. Please define the annotated p values ***/**/* as well as provide the exact p-values for the same in the legend of Figure 1c (a-b), e (a-c); 2c, e, g, i; 3b, e (a, c, e, g); 4c-d; 5c (a-b), e(a-c); 7c (b-c), e-f, h; 8b, c(c), d(d), f, h; 9b, e; EV 2a(b), b(c-d), c(c-d); EV 3b; as appropriate.
 5. Please note that the exact p values are not provided in the legend of Figure 6e.
 6. Please indicate the statistical test used for data analysis in the legends of Figures 1c (a-b), e (a-c); 2c, e, g, i; 3b; 4c-d; 5c (a-b); 7b (b-c), e-f, h; 8b, d(d), f, h; 9b, e; EV 2a(b), b(c-d), c(c-d); EV 3b-c; EV 4c.

7. Please note that the yellow arrowheads are not defined in the legend of Figure 3e (b, d, f, h); 8c (b). This needs to be rectified.

-The section order of the manuscript should be corrected as follows: title page with complete author information, abstract, keywords, introduction, results, discussion, methods, data availability section, acknowledgements, disclosure and competing interests statement, references, main figure legends, tables, expanded figure legends.

Please also note that as part of the EMBO publications' Transparent Editorial Process, The EMBO Journal publishes online a Peer Review File along with each accepted manuscript. This File will be published in conjunction with your paper and will include the referee reports, your point-by-point response and all pertinent correspondence relating to the manuscript. You can opt out of this by letting the editorial office know (contact@embojournal.org). If you do opt out, the Peer Review File link will point to the following statement: "No Peer Review File is available with this article, as the authors have chosen not to make the review process public in this case."

We look forward to seeing a final version of your manuscript as soon as possible. Please let us know if you have any questions and use this link to submit your revision: *Link unavailable*

Best wishes,

Ioannis

Arbitrator:

The authors have fully addressed our concerns: They confirmed that p18 OE actually promotes the transition from NSCs to OB neuron production (increased %Sp8+/RFP+ cells) via astrocytes (Fig9D,E)

We suggest the title to be modified as follows: p18 PROMOTES lineage transitions of excitatory neurons, astrocytes, and interneurons in the MOUSE cortex

All editorial and formatting issues were resolved by the authors.

Dear Yoichi,

Congratulations on an excellent manuscript! I am very pleased to inform you that it has been accepted for publication in The EMBO Journal. Thank you very much for your comprehensive responses to the concerns previously raised by the referees and the arbitrator, and for addressing all our editorial and formatting requests.

If you have any questions, please do not hesitate to contact the Editorial Office. Thank you for your contribution to The EMBO Journal. Working with you has been a pleasure!

Best wishes,

Ioannis
